# A Universal Catalyst for First-Order Optimization

**Hongzhou Lin**[1], **Julien Mairal**[1] **and Zaid Harchaoui**[1,2]
[1]Inria    [2]NYU
{hongzhou.lin,julien.mairal}@inria.fr
zaid.harchaoui@nyu.edu

## Abstract

We introduce a generic scheme for accelerating first-order optimization methods in the sense of Nesterov, which builds upon a new analysis of the accelerated proximal point algorithm. Our approach consists of minimizing a convex objective by approximately solving a sequence of well-chosen auxiliary problems, leading to faster convergence. This strategy applies to a large class of algorithms, including gradient descent, block coordinate descent, SAG, SAGA, SDCA, SVRG, Finito/MISO, and their proximal variants. For all of these methods, we provide acceleration and explicit support for non-strongly convex objectives. In addition to theoretical speed-up, we also show that acceleration is useful in practice, especially for ill-conditioned problems where we measure significant improvements.

## 1   Introduction

A large number of machine learning and signal processing problems are formulated as the minimization of a composite objective function $F : \mathbb{R}^p \rightarrow \mathbb{R}$:

$$\min_{x \in \mathbb{R}^p} \left\{ F(x) \triangleq f(x) + \psi(x) \right\}, \tag{1}$$

where $f$ is convex and has Lipschitz continuous derivatives with constant $L$ and $\psi$ is convex but may not be differentiable. The variable $x$ represents model parameters and the role of $f$ is to ensure that the estimated parameters fit some observed data. Specifically, $f$ is often a large sum of functions

$$f(x) \triangleq \frac{1}{n} \sum_{i=1}^{n} f_i(x), \tag{2}$$

and each term $f_i(x)$ measures the fit between $x$ and a data point indexed by $i$. The function $\psi$ in (1) acts as a regularizer; it is typically chosen to be the squared $\ell_2$-norm, which is smooth, or to be a non-differentiable penalty such as the $\ell_1$-norm or another sparsity-inducing norm [2]. Composite minimization also encompasses constrained minimization if we consider extended-valued indicator functions $\psi$ that may take the value $+\infty$ outside of a convex set $\mathcal{C}$ and 0 inside (see [11]).

Our goal is to accelerate gradient-based or *first-order* methods that are designed to solve (1), with a particular focus on large sums of functions (2). By "accelerating", we mean generalizing a mechanism invented by Nesterov [17] that improves the convergence rate of the gradient descent algorithm. More precisely, when $\psi = 0$, gradient descent steps produce iterates $(x_k)_{k \geq 0}$ such that $F(x_k) - F^* = O(1/k)$, where $F^*$ denotes the minimum value of $F$. Furthermore, when the objective $F$ is strongly convex with constant $\mu$, the rate of convergence becomes linear in $O((1 - \mu/L)^k)$. These rates were shown by Nesterov [16] to be suboptimal for the class of first-order methods, and instead optimal rates—$O(1/k^2)$ for the convex case and $O((1 - \sqrt{\mu/L})^k)$ for the $\mu$-strongly convex one—could be obtained by taking gradient steps at well-chosen points. Later, this acceleration technique was extended to deal with non-differentiable regularization functions $\psi$ [4, 19].

For modern machine learning problems involving a large sum of $n$ functions, a recent effort has been devoted to developing fast *incremental* algorithms [6, 7, 14, 24, 25, 27] that can exploit the particular

structure of (2). Unlike full gradient approaches which require computing and averaging $n$ gradients $\nabla f(x) = (1/n) \sum_{i=1}^{n} \nabla f_i(x)$ at every iteration, incremental techniques have a cost per-iteration that is independent of $n$. The price to pay is the need to store a moderate amount of information regarding past iterates, but the benefit is significant in terms of computational complexity.

**Main contributions.** Our main achievement is a *generic acceleration scheme* that applies to a large class of optimization methods. By analogy with substances that increase chemical reaction rates, we call our approach a "catalyst". A method may be accelerated if it has linear convergence rate for strongly convex problems. This is the case for full gradient [4, 19] and block coordinate descent methods [18, 21], which already have well-known accelerated variants. More importantly, it also applies to incremental algorithms such as SAG [24], SAGA [6], Finito/MISO [7, 14], SDCA [25], and SVRG [27]. Whether or not these methods could be accelerated was an important open question. It was only known to be the case for dual coordinate ascent approaches such as SDCA [26] or SDPC [28] for strongly convex objectives. Our work provides a universal positive answer regardless of the strong convexity of the objective, which brings us to our second achievement.

Some approaches such as Finito/MISO, SDCA, or SVRG are only defined for strongly convex objectives. A classical trick to apply them to general convex functions is to add a small regularization $\varepsilon \|x\|^2$ [25]. The drawback of this strategy is that it requires choosing in advance the parameter $\varepsilon$, which is related to the target accuracy. A consequence of our work is to automatically provide a *direct support for non-strongly convex objectives*, thus removing the need of selecting $\varepsilon$ beforehand.

**Other contribution: Proximal MISO.** The approach Finito/MISO, which was proposed in [7] and [14], is an incremental technique for solving smooth unconstrained $\mu$-strongly convex problems when $n$ is larger than a constant $\beta L/\mu$ (with $\beta = 2$ in [14]). In addition to providing acceleration and support for non-strongly convex objectives, we also make the following specific contributions:
- we extend the method and its convergence proof to deal with the composite problem (1);
- we fix the method to remove the "big data condition" $n \geq \beta L/\mu$.

The resulting algorithm can be interpreted as a variant of proximal SDCA [25] with a different step size and a more practical optimality certificate—that is, checking the optimality condition does not require evaluating a dual objective. Our construction is indeed purely *primal*. Neither our proof of convergence nor the algorithm use duality, while SDCA is originally a dual ascent technique.

**Related work.** The catalyst acceleration can be interpreted as a variant of the proximal point algorithm [3, 9], which is a central concept in convex optimization, underlying augmented Lagrangian approaches, and composite minimization schemes [5, 20]. The proximal point algorithm consists of solving (1) by minimizing a sequence of auxiliary problems involving a quadratic regularization term. In general, these auxiliary problems cannot be solved with perfect accuracy, and several notations of inexactness were proposed, including [9, 10, 22]. The catalyst approach hinges upon (i) an acceleration technique for the proximal point algorithm originally introduced in the pioneer work [9]; (ii) a more practical inexactness criterion than those proposed in the past.[1] As a result, we are able to control the rate of convergence for approximately solving the auxiliary problems with an optimization method $\mathcal{M}$. In turn, we are also able to obtain the computational complexity of the global procedure for solving (1), which was not possible with previous analysis [9, 10, 22]. When instantiated in different first-order optimization settings, our analysis yields systematic acceleration.

Beyond [9], several works have inspired this paper. In particular, accelerated SDCA [26] is an instance of an inexact accelerated proximal point algorithm, even though this was not explicitly stated in [26]. Their proof of convergence relies on different tools than ours. Specifically, we use the concept of *estimate sequence* from Nesterov [17], whereas the direct proof of [26], in the context of SDCA, does not extend to non-strongly convex objectives. Nevertheless, part of their analysis proves to be helpful to obtain our main results. Another useful methodological contribution was the convergence analysis of inexact proximal gradient methods of [23]. Finally, similar ideas appear in the independent work [8]. Their results overlap in part with ours, but both papers adopt different directions. Our analysis is for instance more general and provides support for non-strongly convex objectives. Another independent work with related results is [13], which introduce an accelerated method for the minimization of finite sums, which is not based on the proximal point algorithm.

## 2  The Catalyst Acceleration

We present here our generic acceleration scheme, which can operate on any first-order or gradient-based optimization algorithm with linear convergence rate for strongly convex objectives.

**Linear convergence and acceleration.**  Consider the problem (1) with a $\mu$-strongly convex function $F$, where the strong convexity is defined with respect to the $\ell_2$-norm. A minimization algorithm $\mathcal{M}$, generating the sequence of iterates $(x_k)_{k \geq 0}$, has a *linear convergence rate* if there exists $\tau_{\mathcal{M},F}$ in $(0,1)$ and a constant $C_{\mathcal{M},F}$ in $\mathbb{R}$ such that

$$F(x_k) - F^* \leq C_{\mathcal{M},F}(1 - \tau_{\mathcal{M},F})^k, \tag{3}$$

where $F^*$ denotes the minimum value of $F$. The quantity $\tau_{\mathcal{M},F}$ controls the convergence rate: the larger is $\tau_{\mathcal{M},F}$, the faster is convergence to $F^*$. However, for a given algorithm $\mathcal{M}$, the quantity $\tau_{\mathcal{M},F}$ depends usually on the ratio $L/\mu$, which is often called the *condition number* of $F$.

The catalyst acceleration is a general approach that allows to wrap algorithm $\mathcal{M}$ into an accelerated algorithm $\mathcal{A}$, which enjoys a faster linear convergence rate, with $\tau_{\mathcal{A},F} \geq \tau_{\mathcal{M},F}$. As we will also see, the catalyst acceleration may also be useful when $F$ is not strongly convex—that is, when $\mu = 0$. In that case, we may even consider a method $\mathcal{M}$ that requires strong convexity to operate, and obtain an accelerated algorithm $\mathcal{A}$ that can minimize $F$ with near-optimal convergence rate $\tilde{O}(1/k^2)$.[2]

Our approach can accelerate a wide range of first-order optimization algorithms, starting from classical gradient descent. It also applies to randomized algorithms such as SAG, SAGA, SDCA, SVRG and Finito/MISO, whose rates of convergence are given in expectation. Such methods should be contrasted with stochastic gradient methods [15, 12], which minimize a different non-deterministic function. Acceleration of stochastic gradient methods is beyond the scope of this work.

**Catalyst action.**  We now highlight the mechanics of the catalyst algorithm, which is presented in Algorithm 1. It consists of replacing, at iteration $k$, the original objective function $F$ by an auxiliary objective $G_k$, close to $F$ up to a quadratic term:

$$G_k(x) \triangleq F(x) + \frac{\kappa}{2}\|x - y_{k-1}\|^2, \tag{4}$$

where $\kappa$ will be specified later and $y_k$ is obtained by an extrapolation step described in (6). Then, at iteration $k$, the accelerated algorithm $\mathcal{A}$ minimizes $G_k$ up to accuracy $\varepsilon_k$.

Substituting (4) to (1) has two consequences. On the one hand, minimizing (4) only provides an approximation of the solution of (1), unless $\kappa = 0$; on the other hand, the auxiliary objective $G_k$ enjoys a better condition number than the original objective $F$, which makes it easier to minimize. For instance, when $\mathcal{M}$ is the regular gradient descent algorithm with $\psi = 0$, $\mathcal{M}$ has the rate of convergence (3) for minimizing $F$ with $\tau_{\mathcal{M},F} = \mu/L$. However, owing to the additional quadratic term, $G_k$ can be minimized by $\mathcal{M}$ with the rate (3) where $\tau_{\mathcal{M},G_k} = (\mu + \kappa)/(L + \kappa) > \tau_{\mathcal{M},F}$. In practice, there exists an "optimal" choice for $\kappa$, which controls the time required by $\mathcal{M}$ for solving the auxiliary problems (4), and the quality of approximation of $F$ by the functions $G_k$. This choice will be driven by the convergence analysis in Sec. 3.1-3.3; see also Sec. C for special cases.

**Acceleration via extrapolation and inexact minimization.**  Similar to the classical gradient descent scheme of Nesterov [17], Algorithm 1 involves an extrapolation step (6). As a consequence, the solution of the auxiliary problem (5) at iteration $k+1$ is driven towards the extrapolated variable $y_k$. As shown in [9], this step is in fact sufficient to reduce the number of iterations of Algorithm 1 to solve (1) when $\varepsilon_k = 0$—that is, for running the *exact* accelerated proximal point algorithm.

Nevertheless, to control the total computational complexity of an accelerated algorithm $\mathcal{A}$, it is necessary to take into account the complexity of solving the auxiliary problems (5) using $\mathcal{M}$. This is where our approach differs from the classical proximal point algorithm of [9]. Essentially, both algorithms are the same, but we use the weaker inexactness criterion $G_k(x_k) - G_k^* \leq \varepsilon_k$, where the sequence $(\varepsilon_k)_{k \geq 0}$ is fixed beforehand, and only depends on the initial point. This subtle difference has important consequences: (i) in practice, this condition can often be checked by computing duality gaps; (ii) in theory, the methods $\mathcal{M}$ we consider have linear convergence rates, which allows us to control the complexity of step (5), and then to provide the computational complexity of $\mathcal{A}$.

**Algorithm 1** Catalyst

**input** initial estimate $x_0 \in \mathbb{R}^p$, parameters $\kappa$ and $\alpha_0$, sequence $(\varepsilon_k)_{k \geq 0}$, optimization method $\mathcal{M}$;
  1: Initialize $q = \mu/(\mu + \kappa)$ and $y_0 = x_0$;
  2: **while** the desired stopping criterion is not satisfied **do**
  3:    Find an approximate solution of the following problem using $\mathcal{M}$

$$x_k \approx \underset{x \in \mathbb{R}^p}{\arg\min} \left\{ G_k(x) \triangleq F(x) + \frac{\kappa}{2}\|x - y_{k-1}\|^2 \right\} \quad \text{such that} \quad G_k(x_k) - G_k^* \leq \varepsilon_k. \quad (5)$$

  4:    Compute $\alpha_k \in (0,1)$ from equation $\alpha_k^2 = (1 - \alpha_k)\alpha_{k-1}^2 + q\alpha_k$;
  5:    Compute

$$y_k = x_k + \beta_k(x_k - x_{k-1}) \quad \text{with} \quad \beta_k = \frac{\alpha_{k-1}(1 - \alpha_{k-1})}{\alpha_{k-1}^2 + \alpha_k}. \quad (6)$$

  6: **end while**
**output** $x_k$ (final estimate).

## 3 Convergence Analysis

In this section, we present the theoretical properties of Algorithm 1, for optimization methods $\mathcal{M}$ with deterministic convergence rates of the form (3). When the rate is given as an expectation, a simple extension of our analysis described in Section 4 is needed. For space limitation reasons, we shall sketch the proof mechanics here, and defer the full proofs to Appendix B.

### 3.1 Analysis for $\mu$-Strongly Convex Objective Functions

We first analyze the convergence rate of Algorithm 1 for solving problem 1, regardless of the complexity required to solve the subproblems (5). We start with the $\mu$-strongly convex case.

**Theorem 3.1 (Convergence of Algorithm 1, $\mu$-Strongly Convex Case).**
*Choose $\alpha_0 = \sqrt{q}$ with $q = \mu/(\mu + \kappa)$ and*

$$\varepsilon_k = \frac{2}{9}(F(x_0) - F^*)(1 - \rho)^k \quad \text{with} \quad \rho < \sqrt{q}.$$

*Then, Algorithm 1 generates iterates $(x_k)_{k \geq 0}$ such that*

$$F(x_k) - F^* \leq C(1 - \rho)^{k+1}(F(x_0) - F^*) \quad \text{with} \quad C = \frac{8}{(\sqrt{q} - \rho)^2}. \quad (7)$$

This theorem characterizes the linear convergence rate of Algorithm 1. It is worth noting that the choice of $\rho$ is left to the discretion of the user, but it can safely be set to $\rho = 0.9\sqrt{q}$ in practice. The choice $\alpha_0 = \sqrt{q}$ was made for convenience purposes since it leads to a simplified analysis, but larger values are also acceptable, both from theoretical and practical point of views. Following an advice from Nesterov[17, page 81] originally dedicated to his classical gradient descent algorithm, we may for instance recommend choosing $\alpha_0$ such that $\alpha_0^2 + (1 - q)\alpha_0 - 1 = 0$.

The choice of the sequence $(\varepsilon_k)_{k \geq 0}$ is also subject to discussion since the quantity $F(x_0) - F^*$ is unknown beforehand. Nevertheless, an upper bound may be used instead, which will only affects the corresponding constant in (7). Such upper bounds can typically be obtained by computing a duality gap at $x_0$, or by using additional knowledge about the objective. For instance, when $F$ is non-negative, we may simply choose $\varepsilon_k = (2/9)F(x_0)(1 - \rho)^k$.

The proof of convergence uses the concept of estimate sequence invented by Nesterov [17], and introduces an extension to deal with the errors $(\varepsilon_k)_{k \geq 0}$. To control the accumulation of errors, we borrow the methodology of [23] for inexact proximal gradient algorithms. Our construction yields a convergence result that encompasses both strongly convex and non-strongly convex cases. Note that estimate sequences were also used in [9], but, as noted by [22], the proof of [9] only applies when using an extrapolation step (6) that involves the true minimizer of (5), which is unknown in practice. To obtain a rigorous convergence result like (7), a different approach was needed.

Theorem 3.1 is important, but it does not provide yet the global computational complexity of the full algorithm, which includes the number of iterations performed by $\mathcal{M}$ for approximately solving the auxiliary problems (5). The next proposition characterizes the complexity of this inner-loop.

**Proposition 3.2** (**Inner-Loop Complexity, $\mu$-Strongly Convex Case**).
*Under the assumptions of Theorem 3.1, let us consider a method $\mathcal{M}$ generating iterates $(z_t)_{t \geq 0}$ for minimizing the function $G_k$ with linear convergence rate of the form*

$$G_k(z_t) - G_k^* \leq A(1 - \tau_{\mathcal{M}})^t (G_k(z_0) - G_k^*). \tag{8}$$

*When $z_0 = x_{k-1}$, the precision $\varepsilon_k$ is reached with a number of iterations $T_{\mathcal{M}} = \tilde{O}(1/\tau_{\mathcal{M}})$, where the notation $\tilde{O}$ hides some universal constants and some logarithmic dependencies in $\mu$ and $\kappa$.*

This proposition is generic since the assumption (8) is relatively standard for gradient-based methods [17]. It may now be used to obtain the global rate of convergence of an accelerated algorithm. By calling $F_s$ the objective function value obtained after performing $s = kT_{\mathcal{M}}$ iterations of the method $\mathcal{M}$, the true convergence rate of the accelerated algorithm $\mathcal{A}$ is

$$F_s - F^* = F\left(x_{\frac{s}{T_{\mathcal{M}}}}\right) - F^* \leq C(1 - \rho)^{\frac{s}{T_{\mathcal{M}}}} (F(x_0) - F^*) \leq C\left(1 - \frac{\rho}{T_{\mathcal{M}}}\right)^s (F(x_0) - F^*). \tag{9}$$

As a result, algorithm $\mathcal{A}$ has a global linear rate of convergence with parameter

$$\tau_{\mathcal{A}, F} = \rho/T_{\mathcal{M}} = \tilde{O}(\tau_{\mathcal{M}}\sqrt{\mu}/\sqrt{\mu + \kappa}),$$

where $\tau_{\mathcal{M}}$ typically depends on $\kappa$ (the greater, the faster is $\mathcal{M}$). Consequently, $\kappa$ will be chosen to maximize the ratio $\tau_{\mathcal{M}}/\sqrt{\mu + \kappa}$. Note that for other algorithms $\mathcal{M}$ that do not satisfy (8), additional analysis and possibly a different initialization $z_0$ may be necessary (see Appendix D for example).

## 3.2 Convergence Analysis for Convex but Non-Strongly Convex Objective Functions

We now state the convergence rate when the objective is *not strongly convex*, that is when $\mu = 0$.

**Theorem 3.3** (**Convergence of Algorithm 1, Convex, but Non-Strongly Convex Case**).
*When $\mu = 0$, choose $\alpha_0 = (\sqrt{5} - 1)/2$ and*

$$\varepsilon_k = \frac{2(F(x_0) - F^*)}{9(k+2)^{4+\eta}} \quad \text{with } \eta > 0. \tag{10}$$

*Then, Algorithm 1 generates iterates $(x_k)_{k \geq 0}$ such that*

$$F(x_k) - F^* \leq \frac{8}{(k+2)^2}\left(\left(1 + \frac{2}{\eta}\right)^2 (F(x_0) - F^*) + \frac{\kappa}{2}\|x_0 - x^*\|^2\right). \tag{11}$$

This theorem is the counter-part of Theorem 3.1 when $\mu = 0$. The choice of $\eta$ is left to the discretion of the user; it empirically seem to have very low influence on the global convergence speed, as long as it is chosen small enough (e.g., we use $\eta = 0.1$ in practice). It shows that Algorithm 1 achieves the optimal rate of convergence of first-order methods, but *it does not take into account the complexity of solving the subproblems (5)*. Therefore, we need the following proposition:

**Proposition 3.4** (**Inner-Loop Complexity, Non-Strongly Convex Case**).
*Assume that $F$ has bounded level sets. Under the assumptions of Theorem 3.3, let us consider a method $\mathcal{M}$ generating iterates $(z_t)_{t \geq 0}$ for minimizing the function $G_k$ with linear convergence rate of the form (8). Then, there exists $T_{\mathcal{M}} = \tilde{O}(1/\tau_{\mathcal{M}})$, such that for any $k \geq 1$, solving $G_k$ with initial point $x_{k-1}$ requires at most $T_{\mathcal{M}} \log(k+2)$ iterations of $\mathcal{M}$.*

We can now draw up the global complexity of an accelerated algorithm $\mathcal{A}$ when $\mathcal{M}$ has a linear convergence rate (8) for $\kappa$-strongly convex objectives. To produce $x_k$, $\mathcal{M}$ is called at most $kT_{\mathcal{M}} \log(k+2)$ times. Using the global iteration counter $s = kT_{\mathcal{M}} \log(k+2)$, we get

$$F_s - F^* \leq \frac{8T_{\mathcal{M}}^2 \log^2(s)}{s^2}\left(\left(1 + \frac{2}{\eta}\right)^2 (F(x_0) - F^*) + \frac{\kappa}{2}\|x_0 - x^*\|^2\right). \tag{12}$$

If $\mathcal{M}$ is a first-order method, this rate is *near-optimal*, up to a logarithmic factor, when compared to the optimal rate $O(1/s^2)$, which may be the price to pay for using a generic acceleration scheme.

# 4    Acceleration in Practice

We show here how to accelerate existing algorithms $\mathcal{M}$ and compare the convergence rates obtained before and after catalyst acceleration. For all the algorithms we consider, we study rates of convergence in terms of *total number of iterations* (in expectation, when necessary) to reach accuracy $\varepsilon$. We first show how to accelerate full gradient and randomized coordinate descent algorithms [21]. Then, we discuss other approaches such as SAG [24], SAGA [6], or SVRG [27]. Finally, we present a new proximal version of the incremental gradient approaches Finito/MISO [7, 14], along with its accelerated version. Table 4.1 summarizes the acceleration obtained for the algorithms considered.

**Deriving the global rate of convergence.**    The convergence rate of an accelerated algorithm $\mathcal{A}$ is driven by the parameter $\kappa$. In the strongly convex case, the best choice is the one that maximizes the ratio $\tau_{\mathcal{M},G_k}/\sqrt{\mu + \kappa}$. As discussed in Appendix C, this rule also holds when (8) is given in expectation and in many cases where the constant $\mathcal{C}_{\mathcal{M},G_k}$ is different than $A(G_k(z_0) - G_k^*)$ from (8). When $\mu = 0$, the choice of $\kappa > 0$ only affects the complexity by a multiplicative constant. A rule of thumb is to maximize the ratio $\tau_{\mathcal{M},G_k}/\sqrt{L + \kappa}$ (see Appendix C for more details).

After choosing $\kappa$, the global iteration-complexity is given by Comp $\leq k_{\text{in}} k_{\text{out}}$, where $k_{\text{in}}$ is an upper-bound on the number of iterations performed by $\mathcal{M}$ per inner-loop, and $k_{\text{out}}$ is the upper-bound on the number of outer-loop iterations, following from Theorems 3.1-3.3. Note that for simplicity, we always consider that $L \gg \mu$ such that we may write $L - \mu$ simply as "$L$" in the convergence rates.

## 4.1    Acceleration of Existing Algorithms

**Composite minimization.**    Most of the algorithms we consider here, namely the proximal gradient method [4, 19], SAGA [6], (Prox)-SVRG [27], can handle composite objectives with a regularization penalty $\psi$ that admits a proximal operator $\text{prox}_\psi$, defined for any $z$ as

$$\text{prox}_\psi(z) \triangleq \arg\min_{y \in \mathbb{R}^p} \left\{ \psi(y) + \frac{1}{2}\|y - z\|^2 \right\} .$$

Table 4.1 presents convergence rates that are valid for proximal and non-proximal settings, since most methods we consider are able to deal with such non-differentiable penalties. The exception is SAG [24], for which proximal variants are not analyzed. The incremental method Finito/MISO has also been limited to non-proximal settings so far. In Section 4.2, we actually introduce the extension of MISO to composite minimization, and establish its theoretical convergence rates.

**Full gradient method.**    A first illustration is the algorithm obtained when accelerating the regular "full" gradient descent (FG), and how it contrasts with Nesterov's accelerated variant (AFG). Here, the optimal choice for $\kappa$ is $L - 2\mu$. In the strongly convex case, we get an accelerated rate of convergence in $\tilde{O}(n\sqrt{L/\mu}\log(1/\varepsilon))$, which is the same as AFG up to logarithmic terms. A similar result can also be obtained for randomized coordinate descent methods [21].

**Randomized incremental gradient.**    We now consider randomized incremental gradient methods, resp. SAG [24] and SAGA [6]. When $\mu > 0$, we focus on the "ill-conditioned" setting $n \leq L/\mu$, where these methods have the complexity $O((L/\mu)\log(1/\varepsilon))$. Otherwise, their complexity becomes $O(n\log(1/\varepsilon))$, which is independent of the condition number and seems theoretically optimal [1].

For these methods, the best choice for $\kappa$ has the form $\kappa = a(L - \mu)/(n + b) - \mu$, with $(a, b) = (2, -2)$ for SAG, $(a, b) = (1/2, 1/2)$ for SAGA. A similar formula, with a constant $L'$ in place of $L$, holds for SVRG; we omit it here for brevity. SDCA [26] and Finito/MISO [7, 14] are actually related to incremental gradient methods, and the choice for $\kappa$ has a similar form with $(a, b) = (1, 1)$.

## 4.2    Proximal MISO and its Acceleration

Finito/MISO was proposed in [7] and [14] for solving the problem (1) when $\psi = 0$ and when $f$ is a sum of $n$ $\mu$-strongly convex functions $f_i$ as in (2), which are also differentiable with $L$-Lipschitz derivatives. The algorithm maintains a list of quadratic lower bounds—say $(d_i^k)_{i=1}^n$ at iteration $k$— of the functions $f_i$ and randomly updates one of them at each iteration by using strong-convexity

|  | Comp. $\mu > 0$ | Comp. $\mu = 0$ | Catalyst $\mu > 0$ | Catalyst $\mu = 0$ |
|---|---|---|---|---|
| FG | $O\left(n\left(\frac{L}{\mu}\right)\log\left(\frac{1}{\varepsilon}\right)\right)$ | $O\left(n\frac{L}{\varepsilon}\right)$ | $\tilde{O}\left(n\sqrt{\frac{L}{\mu}}\log\left(\frac{1}{\varepsilon}\right)\right)$ | $\tilde{O}\left(n\frac{L}{\sqrt{\varepsilon}}\right)$ |
| SAG [24] | $O\left(\frac{L}{\mu}\log\left(\frac{1}{\varepsilon}\right)\right)$ | | $\tilde{O}\left(\sqrt{\frac{nL}{\mu}}\log\left(\frac{1}{\varepsilon}\right)\right)$ | |
| SAGA [6] | | | | |
| Finito/MISO-Prox | | | | |
| SDCA [25] | | not avail. | | |
| SVRG [27] | $O\left(\frac{L'}{\mu}\log\left(\frac{1}{\varepsilon}\right)\right)$ | | $\tilde{O}\left(\sqrt{\frac{nL'}{\mu}}\log\left(\frac{1}{\varepsilon}\right)\right)$ | |
| Acc-FG [19] | $O\left(n\sqrt{\frac{L}{\mu}}\log\left(\frac{1}{\varepsilon}\right)\right)$ | $O\left(n\frac{L}{\sqrt{\varepsilon}}\right)$ | no acceleration | |
| Acc-SDCA [26] | $\tilde{O}\left(\sqrt{\frac{nL}{\mu}}\log\left(\frac{1}{\varepsilon}\right)\right)$ | not avail. | | |

Table 1: Comparison of rates of convergence, before and after the catalyst acceleration, resp. in the strongly-convex and non strongly-convex cases. **To simplify, we only present the case where $n \leq L/\mu$ when $\mu > 0$.** For all incremental algorithms, there is indeed no acceleration otherwise. The quantity $L'$ for SVRG is the average Lipschitz constant of the functions $f_i$ (see [27]).

inequalities. The current iterate $x_k$ is then obtained by minimizing the lower-bound of the objective

$$x_k = \arg\min_{x \in \mathbb{R}^p} \left\{ D_k(x) = \frac{1}{n} \sum_{i=1}^{n} d_i^k(x) \right\}. \tag{13}$$

Interestingly, since $D_k$ is a lower-bound of $F$ we also have $D_k(x_k) \leq F^*$, and thus the quantity $F(x_k) - D_k(x_k)$ can be used as an optimality certificate that upper-bounds $F(x_k) - F^*$. Furthermore, this certificate was shown to converge to zero with a rate similar to SAG/SDCA/SVRG/SAGA under the condition $n \geq 2L/\mu$. In this section, we show how to remove this condition and how to provide support to non-differentiable functions $\psi$ whose proximal operator can be easily computed. We shall briefly sketch the main ideas, and we refer to Appendix D for a thorough presentation.

The first idea to deal with a nonsmooth regularizer $\psi$ is to change the definition of $D_k$:

$$D_k(x) = \frac{1}{n} \sum_{i=1}^{n} d_i^k(x) + \psi(x),$$

which was also proposed in [7] without a convergence proof. Then, because the $d_i^k$'s are quadratic functions, the minimizer $x_k$ of $D_k$ can be obtained by computing the proximal operator of $\psi$ at a particular point. The second idea to remove the condition $n \geq 2L/\mu$ is to modify the update of the lower bounds $d_i^k$. Assume that index $i_k$ is selected among $\{1, \ldots, n\}$ at iteration $k$, then

$$d_i^k(x) = \begin{cases} (1-\delta)d_i^{k-1}(x) + \delta(f_i(x_{k-1}) + \langle \nabla f_i(x_{k-1}), x - x_{k-1}\rangle + \frac{\mu}{2}\|x - x_{k-1}\|^2) & \text{if } i = i_k \\ d_i^{k-1}(x) & \text{otherwise} \end{cases}$$

Whereas the original Finito/MISO uses $\delta = 1$, our new variant uses $\delta = \min(1, \mu n/2(L-\mu))$. The resulting algorithm turns out to be very close to variant "5" of proximal SDCA [25], which corresponds to using a different value for $\delta$. The main difference between SDCA and MISO-Prox is that the latter does not use duality. It also provides a different (simpler) optimality certificate $F(x_k) - D_k(x_k)$, which is guaranteed to converge linearly, as stated in the next theorem.

**Theorem 4.1 (Convergence of MISO-Prox).**
*Let $(x_k)_{k \geq 0}$ be obtained by MISO-Prox, then*

$$\mathbb{E}[F(x_k)] - F^* \leq \frac{1}{\tau}(1-\tau)^{k+1}\left(F(x_0) - D_0(x_0)\right) \quad \text{with } \tau \geq \min\left\{\frac{\mu}{4L}, \frac{1}{2n}\right\}. \tag{14}$$

*Furthermore, we also have fast convergence of the certificate*

$$\mathbb{E}[F(x_k) - D_k(x_k)] \leq \frac{1}{\tau}(1-\tau)^k \left(F^* - D_0(x_0)\right).$$

The proof of convergence is given in Appendix D. Finally, we conclude this section by noting that MISO-Prox enjoys the catalyst acceleration, leading to the iteration-complexity presented in Table 4.1. Since the convergence rate (14) does not have exactly the same form as (8), Propositions 3.2 and 3.4 cannot be used and additional analysis, given in Appendix D, is needed. Practical forms of the algorithm are also presented there, along with discussions on how to initialize it.

# 5 Experiments

We evaluate the Catalyst acceleration on three methods that have never been accelerated in the past: SAG [24], SAGA [6], and MISO-Prox. We focus on $\ell_2$-regularized logistic regression, where the regularization parameter $\mu$ yields a lower bound on the strong convexity parameter of the problem.

We use three datasets used in [14], namely real-sim, rcv1, and ocr, which are relatively large, with up to $n = 2\,500\,000$ points for ocr and $p = 47\,152$ variables for rcv1. We consider three regimes: $\mu = 0$ (no regularization), $\mu/L = 0.001/n$ and $\mu/L = 0.1/n$, which leads significantly larger condition numbers than those used in other studies ($\mu/L \approx 1/n$ in [14, 24]). We compare MISO, SAG, and SAGA with their default parameters, which are recommended by their theoretical analysis (step-sizes $1/L$ for SAG and $1/3L$ for SAGA), and study several accelerated variants. The values of $\kappa$ and $\rho$ and the sequences $(\varepsilon_k)_{k \geq 0}$ are those suggested in the previous sections, with $\eta = 0.1$ in (10). Other implementation details are presented in Appendix E.

The restarting strategy for $\mathcal{M}$ is key to achieve acceleration in practice. All of the methods we compare store $n$ gradients evaluated at previous iterates of the algorithm. We always use the gradients from the previous run of $\mathcal{M}$ to initialize a new one. We detail in Appendix E the initialization for each method. Finally, we evaluated a heuristic that constrain $\mathcal{M}$ to always perform at most $n$ iterations (one pass over the data); we call this variant AMISO2 for MISO whereas AMISO1 refers to the regular "vanilla" accelerated variant, and we also use this heuristic to accelerate SAG.

The results are reported in Table 1. We always obtain a huge speed-up for MISO, which suffers from numerical stability issues when the condition number is very large (for instance, $\mu/L = 10^{-3}/n = 4.10^{-10}$ for ocr). Here, not only does the catalyst algorithm accelerate MISO, but it also stabilizes it. Whereas MISO is slower than SAG and SAGA in this "small $\mu$" regime, AMISO2 is almost systematically the best performer. We are also able to accelerate SAG and SAGA in general, even though the improvement is less significant than for MISO. In particular, SAGA without acceleration proves to be the best method on ocr. One reason may be its ability to adapt to the unknown strong convexity parameter $\mu' \geq \mu$ of the objective near the solution. When $\mu'/L \geq 1/n$, we indeed obtain a regime where acceleration does not occur (see Sec. 4). Therefore, this experiment suggests that adaptivity to unknown strong convexity is of high interest for incremental optimization.

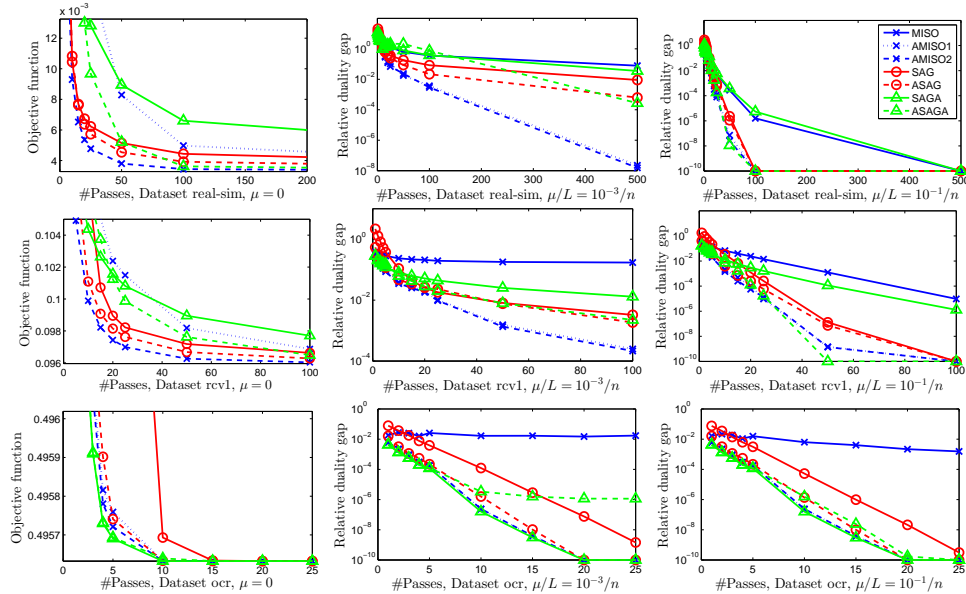

Figure 1: Objective function value (or duality gap) for different number of passes performed over each dataset. The legend for all curves is on the top right. AMISO, ASAGA, ASAG refer to the accelerated variants of MISO, SAGA, and SAG, respectively.

### Acknowledgments

This work was supported by ANR (MACARON ANR-14-CE23-0003-01), MSR-Inria joint centre, CNRS-Mastodons program (Titan), and NYU Moore-Sloan Data Science Environment.

## Footnotes

[1]Note that our inexact criterion was also studied, among others, in [22], but the analysis of [22] led to the conjecture that this criterion was too weak to warrant acceleration. Our analysis refutes this conjecture.

[2]In this paper, we use the notation $O(.)$ to hide constants. The notation $\tilde{O}(.)$ also hides logarithmic factors.

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
