[Supplementary Material]

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

[3]Note that even though we call this algorithm MISO (or Finito), it was called MISO$\mu$ in [14], whereas "MISO" was originally referring to an incremental majorization-minimization procedure that uses upper bounds of the functions $f_i$ instead of lower bounds, which is appropriate for non-convex optimization problems.

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

In this appendix, Section A is devoted to the construction of an object called *estimate sequence*, originally introduced by Nesterov (see [17]), and introduce extensions to deal with inexact minimization. This section contains a generic convergence result that will be used to prove the main theorems and propositions of the paper in Section B. Then, Section C is devoted to the computation of global convergence rates of accelerated algorithms, Section D presents in details the proximal MISO algorithm, and Section E gives some implementation details of the experiments.

## A    Construction of the Approximate Estimate Sequence

The estimate sequence is a generic tool introduced by Nesterov for proving the convergence of accelerated gradient-based algorithms. We start by recalling the definition given in [17].

**Definition A.1 (Estimate Sequence [17]).**
A pair of sequences $(\varphi_k)_{k \geq 0}$ and $(\lambda_k)_{k \geq 0}$, with $\lambda_k \geq 0$ and $\varphi_k : \mathbb{R}^p \to \mathbb{R}$, is called an *estimate sequence* of function $F$ if

$$\lambda_k \to 0,$$

and for any $x$ in $\mathbb{R}^p$ and all $k \geq 0$, we have

$$\varphi_k(x) \leq (1 - \lambda_k)F(x) + \lambda_k \varphi_0(x).$$

Estimate sequences are used for proving convergence rates thanks to the following lemma

**Lemma A.2 (Lemma 2.2.1 from [17]).**
*If for some sequence $(x_k)_{k \geq 0}$ we have*

$$F(x_k) \leq \varphi_k^* \triangleq \min_{x \in \mathbb{R}^p} \varphi_k(x),$$

*for an estimate sequence $(\varphi_k)_{k \geq 0}$ of $F$, then*

$$F(x_k) - F^* \leq \lambda_k(\varphi_0(x^*) - F^*),$$

*where $x^*$ is a minimizer of $F$.*

The rate of convergence of $F(x_k)$ is thus directly related to the convergence rate of $\lambda_k$. Constructing estimate sequences is thus appealing, even though finding the most appropriate one is not trivial for the catalyst algorithm because of the approximate minimization of $G_k$ in (5). In a nutshell, the main steps of our convergence analysis are

1. define an "approximate" estimate sequence for $F$ corresponding to Algorithm 1—that is, finding a function $\varphi$ that almost satisfies Definition A.1 up to the approximation errors $\varepsilon_k$ made in (5) when minimizing $G_k$, and control the way these errors sum up together.

2. extend Lemma A.2 to deal with the approximation errors $\varepsilon_k$ to derive a generic convergence rate for the sequence $(x_k)_{k \geq 0}$.

This is also the strategy proposed by Güler in [9] for his inexact accelerated proximal point algorithm, which essentially differs from ours in its stopping criterion. The estimate sequence we choose is also different and leads to a more rigorous convergence proof. Specifically, we prove in this section the following theorem:

**Theorem A.3 (Convergence Result Derived from an Approximate Estimate Sequence).**
*Let us denote*

$$\lambda_k = \prod_{i=0}^{k-1}(1 - \alpha_i), \tag{15}$$

*where the $\alpha_i$'s are defined in Algorithm 1. Then, the sequence $(x_k)_{k \geq 0}$ satisfies*

$$F(x_k) - F^* \leq \lambda_k \left( \sqrt{S_k} + 2 \sum_{i=1}^{k} \sqrt{\frac{\varepsilon_i}{\lambda_i}} \right)^2, \tag{16}$$

*where $F^*$ is the minimum value of $F$ and*

$$S_k = F(x_0) - F^* + \frac{\gamma_0}{2}\|x_0 - x^*\|^2 + \sum_{i=1}^{k} \frac{\varepsilon_i}{\lambda_i} \quad where \quad \gamma_0 = \frac{\alpha_0\left((\kappa + \mu)\alpha_0 - \mu\right)}{1 - \alpha_0}, \tag{17}$$

*where $x^*$ is a minimizer of $F$.*

Then, the theorem will be used with the following lemma from [17] to control the convergence rate of the sequence $(\lambda_k)_{k \geq 0}$, whose definition follows the classical use of estimate sequences [17]. This will provide us convergence rates both for the strongly convex and non-strongly convex cases.

**Lemma A.4 (Lemma 2.2.4 from [17]).**
*If in the quantity $\gamma_0$ defined in (17) satisfies $\gamma_0 \geq \mu$, then the sequence $(\lambda_k)_{k \geq 0}$ from (15) satisfies*

$$\lambda_k \leq \min \left\{ (1 - \sqrt{q})^k, \frac{4}{\left(2 + k\sqrt{\frac{\gamma_0}{\kappa + \mu}}\right)^2} \right\}. \tag{18}$$

We may now move to the proof of the theorem.

## A.1 Proof of Theorem A.3

The first step is to construct an estimate sequence is typically to find a sequence of lower bounds of $F$. By calling $x_k^*$ the minimizer of $G_k$, the following one is used in [9]:

**Lemma A.5 (Lower Bound for $F$ near $x_k^*$).**
*For all $x$ in $\mathbb{R}^p$,*

$$F(x) \geq F(x_k^*) + \langle \kappa(y_{k-1} - x_k^*), x - x_k^* \rangle + \frac{\mu}{2} \|x - x_k^*\|^2. \tag{19}$$

*Proof.* By strong convexity, $G_k(x) \geq G_k(x_k^*) + \frac{\kappa + \mu}{2} \|x - x_k^*\|^2$, which is equivalent to

$$F(x) + \frac{\kappa}{2} \|x - y_k\|^2 \geq F(x_k^*) + \frac{\kappa}{2} \|x_k^* - y_{k-1}\|^2 + \frac{\kappa + \mu}{2} \|x - x_k^*\|^2.$$

After developing the quadratic terms, we directly obtain (19). $\qquad\square$

Unfortunately, the exact value $x_k^*$ is unknown in practice and the estimate sequence of [9] yields in fact an algorithm where the definition of the anchor point $y_k$ involves the unknown quantity $x_k^*$ instead of the approximate solutions $x_k$ and $x_{k-1}$ as in (6), as also noted by others [22]. To obtain a rigorous proof of convergence for Algorithm 1, it is thus necessary to refine the analysis of [9]. To that effect, we construct below a sequence of functions that approximately satisfies the definition of estimate sequences. Essentially, we replace in (19) the quantity $x_k^*$ by $x_k$ to obtain an approximate lower bound, and control the error by using the condition $G_k(x_k) - G_k(x_k^*) \leq \varepsilon_k$. This leads us to the following construction:

1. $\phi_0(x) = F(x_0) + \frac{\gamma_0}{2} \|x - x_0\|^2$;

2. For $k \geq 1$, we set

$$\phi_k(x) = (1 - \alpha_{k-1})\phi_{k-1}(x) + \alpha_{k-1}[F(x_k) + \langle \kappa(y_{k-1} - x_k), x - x_k \rangle + \frac{\mu}{2} \|x - x_k\|^2],$$

where the value of $\gamma_0$, given in (17) will be explained later. Note that if one replaces $x_k$ by $x_k^*$ in the above construction, it is easy to show that $(\phi_k)_{k \geq 0}$ would be exactly an estimate sequence for $F$ with the relation $\lambda_k$ given in (15).

Before extending Lemma A.2 to deal with the approximate sequence and conclude the proof of the theorem, we need to characterize a few properties of the sequence $(\phi_k)_{k \geq 0}$. In particular, the functions $\phi_k$ are quadratic and admit a canonical form:

**Lemma A.6 (Canonical Form of the Functions $\phi_k$).**
*For all $k \geq 0$, $\phi_k$ can be written in the canonical form*

$$\phi_k(x) = \phi_k^* + \frac{\gamma_k}{2} \|x - v_k\|^2,$$

*where the sequences* $(\gamma_k)_{k \geq 0}$, $(v_k)_{k \geq 0}$, *and* $(\phi_k^*)_{k \geq 0}$ *are defined as follows*

$$\gamma_k = (1 - \alpha_{k-1})\gamma_{k-1} + \alpha_{k-1}\mu, \tag{20}$$

$$v_k = \frac{1}{\gamma_k}\left((1 - \alpha_{k-1})\gamma_{k-1}v_{k-1} + \alpha_{k-1}\mu x_k - \alpha_{k-1}\kappa(y_{k-1} - x_k)\right), \tag{21}$$

$$\phi_k^* = (1 - \alpha_{k-1})\phi_{k-1}^* + \alpha_{k-1}F(x_k) - \frac{\alpha_{k-1}^2}{2\gamma_k}\|\kappa(y_{k-1} - x_k)\|^2$$
$$+ \frac{\alpha_{k-1}(1 - \alpha_{k-1})\gamma_{k-1}}{\gamma_k}\left(\frac{\mu}{2}\|x_k - v_{k-1}\|^2 + \langle \kappa(y_{k-1} - x_k), v_{k-1} - x_k\rangle\right), \tag{22}$$

*Proof.* We have for all $k \geq 1$ and all $x$ in $\mathbb{R}^p$,

$$\phi_k(x) = (1 - \alpha_{k-1})\left(\phi_{k-1}^* + \frac{\gamma_{k-1}}{2}\|x - v_{k-1}\|^2\right)$$
$$+ \alpha_{k-1}\left(F(x_k) + \langle \kappa(y_{k-1} - x_k), x - x_k\rangle + \frac{\mu}{2}\|x - x_k\|^2\right) \tag{23}$$
$$= \phi_k^* + \frac{\gamma_k}{2}\|x - v_k\|^2.$$

Differentiate twice the relations (23) gives us directly (20). Since $v_k$ minimizes $\phi_k$, the optimality condition $\nabla\phi_k(v_k) = 0$ gives

$$(1 - \alpha_{k-1})\gamma_{k-1}(v_k - v_{k-1}) + \alpha_{k-1}\left(\kappa(y_{k-1} - x_k) + \mu(v_k - x_k)\right) = 0,$$

and then we obtain (21). Finally, apply $x = x_k$ to (23), which yields

$$\phi_k(x_k) = (1 - \alpha_{k-1})\left(\phi_{k-1}^* + \frac{\gamma_{k-1}}{2}\|x_k - v_{k-1}\|^2\right) + \alpha_{k-1}F(x_k) = \phi_k^* + \frac{\gamma_k}{2}\|x_k - v_k\|^2.$$

Consequently,

$$\phi_k^* = (1 - \alpha_{k-1})\phi_{k-1}^* + \alpha_{k-1}F(x_k) + (1 - \alpha_{k-1})\frac{\gamma_{k-1}}{2}\|x_k - v_{k-1}\|^2 - \frac{\gamma_k}{2}\|x_k - v_k\|^2 \tag{24}$$

Using the expression of $v_k$ from (21), we have

$$v_k - x_k = \frac{1}{\gamma_k}\left((1 - \alpha_{k-1})\gamma_{k-1}(v_{k-1} - x_k) - \alpha_{k-1}\kappa(y_{k-1} - x_k)\right).$$

Therefore

$$\frac{\gamma_k}{2}\|x_k - v_k\|^2 = \frac{(1 - \alpha_{k-1})^2\gamma_{k-1}^2}{2\gamma_k}\|x_k - v_{k-1}\|^2$$
$$- \frac{(1 - \alpha_{k-1})\alpha_{k-1}\gamma_{k-1}}{\gamma_k}\langle v_{k-1} - x_k, \kappa(y_{k-1} - x_k)\rangle + \frac{\alpha_{k-1}^2}{2\gamma_k}\|\kappa(y_{k-1} - x_k)\|^2.$$

It remains to plug this relation into (24), use once (20), and we obtain the formula (22) for $\phi_k^*$. $\qquad\square$

We may now start analyzing the errors $\varepsilon_k$ to control how far is the sequence $(\phi_k)_{k \geq 0}$ from an exact estimate sequence. For that, we need to understand the effect of replacing $x_k^*$ by $x_k$ in the lower bound (19). The following lemma will be useful for that purpose.

**Lemma A.7 (Controlling the Approximate Lower Bound of $F$).**
*If* $G_k(x_k) - G_k(x_k^*) \leq \varepsilon_k$, *then for all* $x$ *in* $\mathbb{R}^p$,

$$F(x) \geq F(x_k) + \langle \kappa(y_{k-1} - x_k), x - x_k\rangle + \frac{\mu}{2}\|x - x_k\|^2 + (\kappa + \mu)\langle x_k - x_k^*, x - x_k\rangle - \varepsilon_k. \tag{25}$$

*Proof.* By strong convexity, for all $x$ in $\mathbb{R}^p$,

$$G_k(x) \geq G_k^* + \frac{\kappa + \mu}{2}\|x - x_k^*\|^2,$$

where $G_k^*$ is the minimum value of $G_k$. Replacing $G_k$ by its definition (5) gives

$$F(x) \geq G_k^* + \frac{\kappa + \mu}{2}\|x - x_k^*\|^2 - \frac{\kappa}{2}\|x - y_{k-1}\|^2$$

$$= G_k(x_k) + (G_k^* - G_k(x_k)) + \frac{\kappa + \mu}{2}\|x - x_k^*\|^2 - \frac{\kappa}{2}\|x - y_{k-1}\|^2$$

$$\geq G_k(x_k) - \varepsilon_k + \frac{\kappa + \mu}{2}\|(x - x_k) + (x_k - x_k^*)\|^2 - \frac{\kappa}{2}\|x - y_{k-1}\|^2$$

$$\geq G_k(x_k) - \varepsilon_k + \frac{\kappa + \mu}{2}\|x - x_k\|^2 - \frac{\kappa}{2}\|x - y_{k-1}\|^2 + (\kappa + \mu)\langle x_k - x_k^*, x - x_k\rangle.$$

We conclude by noting that

$$G_k(x_k) + \frac{\kappa}{2}\|x - x_k\|^2 - \frac{\kappa}{2}\|x - y_{k-1}\|^2 = F(x_k) + \frac{\kappa}{2}\|x_k - y_{k-1}\|^2 + \frac{\kappa}{2}\|x - x_k\|^2 - \frac{\kappa}{2}\|x - y_{k-1}\|^2$$

$$= F(x_k) + \langle \kappa(y_{k-1} - x_k), x - x_k\rangle.$$

$\square$

We can now show that Algorithm 1 generates iterates $(x_k)_{k\geq 0}$ that approximately satisfy the condition of Lemma A.2 from Nesterov [17].

**Lemma A.8** (**Relation between** $(\phi_k)_{k\geq 0}$ **and Algorithm 1**).
*Let $\phi_k$ be the estimate sequence constructed above. Then, Algorithm 1 generates iterates $(x_k)_{k\geq 0}$ such that*

$$F(x_k) \leq \phi_k^* + \xi_k,$$

*where the sequence $(\xi_k)_{k\geq 0}$ is defined by $\xi_0 = 0$ and*

$$\xi_k = (1 - \alpha_{k-1})(\xi_{k-1} + \varepsilon_k - (\kappa + \mu)\langle x_k - x_k^*, x_{k-1} - x_k\rangle).$$

*Proof.* We proceed by induction. For $k = 0$, $\phi_0^* = F(x_0)$ and $\xi_0 = 0$.
Assume now that $F(x_{k-1}) \leq \phi_{k-1}^* + \xi_{k-1}$. Then,

$$\phi_{k-1}^* \geq F(x_{k-1}) - \xi_{k-1}$$

$$\geq F(x_k) + \langle \kappa(y_{k-1} - x_k), x_{k-1} - x_k\rangle + (\kappa + \mu)\langle x_k - x_k^*, x_{k-1} - x_k\rangle - \varepsilon_k - \xi_{k-1}$$

$$= F(x_k) + \langle \kappa(y_{k-1} - x_k), x_{k-1} - x_k\rangle - \xi_k/(1 - \alpha_{k-1}),$$

where the second inequality is due to (25). By Lemma A.6, we now have,

$$\phi_k^* = (1 - \alpha_{k-1})\phi_{k-1}^* + \alpha_{k-1}F(x_k) - \frac{\alpha_{k-1}^2}{2\gamma_k}\|\kappa(y_{k-1} - x_k)\|^2$$

$$+ \frac{\alpha_{k-1}(1 - \alpha_{k-1})\gamma_{k-1}}{\gamma_k}\left(\frac{\mu}{2}\|x_k - v_{k-1}\|^2 + \langle \kappa(y_{k-1} - x_k), v_{k-1} - x_k\rangle\right)$$

$$\geq (1 - \alpha_{k-1})\left(F(x_k) + \langle \kappa(y_{k-1} - x_k), x_{k-1} - x_k\rangle\right) - \xi_k + \alpha_{k-1}F(x_k)$$

$$- \frac{\alpha_{k-1}^2}{2\gamma_k}\|\kappa(y_{k-1} - x_k)\|^2 + \frac{\alpha_{k-1}(1 - \alpha_{k-1})\gamma_{k-1}}{\gamma_k}\langle \kappa(y_{k-1} - x_k), v_{k-1} - x_k\rangle.$$

$$= F(x_k) + (1 - \alpha_{k-1})\langle \kappa(y_{k-1} - x_k), x_{k-1} - x_k + \frac{\alpha_{k-1}\gamma_{k-1}}{\gamma_k}(v_{k-1} - x_k)\rangle$$

$$- \frac{\alpha_{k-1}^2}{2\gamma_k}\|\kappa(y_{k-1} - x_k)\|^2 - \xi_k$$

$$= F(x_k) + (1 - \alpha_{k-1})\langle \kappa(y_{k-1} - x_k), x_{k-1} - y_{k-1} + \frac{\alpha_{k-1}\gamma_{k-1}}{\gamma_k}(v_{k-1} - y_{k-1})\rangle$$

$$+ \left(1 - \frac{(\kappa + 2\mu)\alpha_{k-1}^2}{2\gamma_k}\right)\kappa\|(y_{k-1} - x_k)\|^2 - \xi_k.$$

We now need to show that the choice of the sequences $(\alpha_k)_{k\geq 0}$ and $(y_k)_{k\geq 0}$ will cancel all the terms involving $y_{k-1} - x_k$. In other words, we want to show that

$$x_{k-1} - y_{k-1} + \frac{\alpha_{k-1}\gamma_{k-1}}{\gamma_k}(v_{k-1} - y_{k-1}) = 0, \qquad (26)$$

and we want to show that

$$1 - (\kappa + \mu)\frac{\alpha_{k-1}^2}{\gamma_k} = 0, \tag{27}$$

which will be sufficient to conclude that $\phi_k^* + \xi_k \geq F(x_k)$. The relation (27) can be obtained from the definition of $\alpha_k$ in (6) and the form of $\gamma_k$ given in (20). We have indeed from (6) that

$$(\kappa + \mu)\alpha_k^2 = (1 - \alpha_k)(\kappa + \mu)\alpha_{k-1}^2 + \alpha_k \mu.$$

Then, the quantity $(\kappa + \mu)\alpha_k^2$ follows the same recursion as $\gamma_{k+1}$ in (20). Moreover, we have

$$\gamma_1 = (1 - \alpha_0)\gamma_0 + \mu\alpha_0 = (\kappa + \mu)\alpha_0^2,$$

from the definition of $\gamma_0$ in (17). We can then conclude by induction that $\gamma_{k+1} = (\kappa + \mu)\alpha_k^2$ for all $k \geq 0$ and (27) is satisfied.

To prove (26), we assume that $y_{k-1}$ is chosen such that (26) is satisfied, and show that it is equivalent to defining $y_k$ as in (6). By lemma A.6,

$$\begin{aligned}
v_k &= \frac{1}{\gamma_k}\left((1 - \alpha_{k-1})\gamma_{k-1}v_{k-1} + \alpha_{k-1}\mu x_k - \alpha_{k-1}\kappa(y_{k-1} - x_k)\right) \\
&= \frac{1}{\gamma_k}\left(\frac{(1 - \alpha_{k-1})}{\alpha_{k-1}}((\gamma_k + \alpha_{k-1}\gamma_{k-1})y_{k-1} - \gamma_k x_{k-1}) + \alpha_{k-1}\mu x_k - \alpha_{k-1}\kappa(y_{k-1} - x_k)\right) \\
&= \frac{1}{\gamma_k}\left(\frac{(1 - \alpha_{k-1})}{\alpha_{k-1}}((\gamma_{k-1} + \alpha_{k-1}\mu)y_{k-1} - \gamma_k x_{k-1}) + \alpha_{k-1}(\mu + \kappa)x_k - \alpha_{k-1}\kappa y_{k-1}\right) \\
&= \frac{1}{\gamma_k}\left(\frac{1}{\alpha_{k-1}}(\gamma_k - \mu\alpha_{k-1}^2)y_{k-1} - \frac{(1 - \alpha_{k-1})}{\alpha_{k-1}}\gamma_k x_{k-1} + \frac{\gamma_k}{\alpha_{k-1}}x_k - \alpha_{k-1}\kappa y_{k-1}\right) \\
&= \frac{1}{\alpha_{k-1}}(x_k - (1 - \alpha_{k-1})x_{k-1}), \tag{28}
\end{aligned}$$

As a result, using (26) by replacing $k - 1$ by $k$ yields

$$y_k = x_k + \frac{\alpha_{k-1}(1 - \alpha_{k-1})}{\alpha_{k-1}^2 + \alpha_k}(x_k - x_{k-1}),$$

and we obtain the original equivalent definition of (6). This concludes the proof. □

With this lemma in hand, we introduce the following proposition, which brings us almost to Theorem A.3, which we want to prove.

**Proposition A.9 (Auxiliary Proposition for Theorem A.3).**
*Let us consider the sequence $(\lambda_k)_{k\geq 0}$ defined in (15). Then, the sequence $(x_k)_{k\geq 0}$ satisfies*

$$\frac{1}{\lambda_k}(F(x_k) - F^* + \frac{\gamma_k}{2}\|x^* - v_k\|^2) \leq \phi_0(x^*) - F^* + \sum_{i=1}^{k}\frac{\varepsilon_i}{\lambda_i} + \sum_{i=1}^{k}\frac{\sqrt{2\varepsilon_i\gamma_i}}{\lambda_i}\|x^* - v_i\|,$$

*where $x^*$ is a minimizer of $F$ and $F^*$ its minimum value.*

*Proof.* By the definition of the function $\phi_k$, we have

$$\begin{aligned}
\phi_k(x^*) &= (1 - \alpha_{k-1})\phi_{k-1}(x^*) + \alpha_{k-1}[F(x_k) + \langle \kappa(y_{k-1} - x_k), x^* - x_k\rangle + \frac{\mu}{2}\|x^* - x_k\|^2] \\
&\leq (1 - \alpha_{k-1})\phi_{k-1}(x^*) + \alpha_{k-1}[F(x^*) + \varepsilon_k - (\kappa + \mu)\langle x_k - x_k^*, x^* - x_k\rangle],
\end{aligned}$$

where the inequality comes from (25). Therefore, by using the definition of $\xi_k$ in Lemma A.8,

$$\begin{aligned}
&\phi_k(x^*) + \xi_k - F^* \\
\leq\ & (1 - \alpha_{k-1})(\phi_{k-1}(x^*) + \xi_{k-1} - F^*) + \varepsilon_k - (\kappa + \mu)\langle x_k - x_k^*, (1 - \alpha_{k-1})x_{k-1} + \alpha_{k-1}x^* - x_k\rangle \\
=\ & (1 - \alpha_{k-1})(\phi_{k-1}(x^*) + \xi_{k-1} - F^*) + \varepsilon_k - \alpha_{k-1}(\kappa + \mu)\langle x_k - x_k^*, x^* - v_k\rangle \\
\leq\ & (1 - \alpha_{k-1})(\phi_{k-1}(x^*) + \xi_{k-1} - F^*) + \varepsilon_k + \alpha_{k-1}(\kappa + \mu)\|x_k - x_k^*\|\|x^* - v_k\| \\
\leq\ & (1 - \alpha_{k-1})(\phi_{k-1}(x^*) + \xi_{k-1} - F^*) + \varepsilon_k + \alpha_{k-1}\sqrt{2(\kappa + \mu)\varepsilon_k}\|x^* - v_k\| \\
=\ & (1 - \alpha_{k-1})(\phi_{k-1}(x^*) + \xi_{k-1} - F^*) + \varepsilon_k + \sqrt{2\varepsilon_k\gamma_k}\|x^* - v_k\|,
\end{aligned}$$

where the first equality uses the relation (28), the last inequality comes from the strong convexity relation $\varepsilon_k \geq G_k(x_k) - G_k(x_k^*) \geq (1/2)(\kappa + \mu)\|x_k^* - x_k\|^2$, and the last equality uses the relation $\gamma_k = (\kappa + \mu)\alpha_{k-1}^2$.

Dividing both sides by $\lambda_k$ yields

$$\frac{1}{\lambda_k}(\phi_k(x^*) + \xi_k - F^*) \leq \frac{1}{\lambda_{k-1}}(\phi_{k-1}(x^*) + \xi_{k-1} - F^*) + \frac{\varepsilon_k}{\lambda_k} + \frac{\sqrt{2\varepsilon_k\gamma_k}}{\lambda_k}\|x^* - v_k\|.$$

A simple recurrence gives,

$$\frac{1}{\lambda_k}(\phi_k(x^*) + \xi_k - F^*) \leq \phi_0(x^*) - F^* + \sum_{i=1}^{k}\frac{\varepsilon_i}{\lambda_i} + \sum_{i=1}^{k}\frac{\sqrt{2\varepsilon_i\gamma_i}}{\lambda_i}\|x^* - v_i\|.$$

Finally, by lemmas A.6 and A.8,

$$\phi_k(x^*) + \xi_k - F^* = \frac{\gamma_k}{2}\|x^* - v_k\|^2 + \phi_k^* + \xi_k - F^* \geq \frac{\gamma_k}{2}\|x^* - v_k\|^2 + F(x_k) - F^*.$$

As a result,

$$\frac{1}{\lambda_k}(F(x_k) - F^* + \frac{\gamma_k}{2}\|x^* - v_k\|^2) \leq \phi_0(x^*) - F^* + \sum_{i=1}^{k}\frac{\varepsilon_i}{\lambda_i} + \sum_{i=1}^{k}\frac{\sqrt{2\varepsilon_i\gamma_i}}{\lambda_i}\|x^* - v_i\|. \quad (29)$$

$\square$

To control the error term on the right and finish the proof of Theorem A.3, we are going to borrow some methodology used to analyze the convergence of inexact proximal gradient algorithms from [23], and use an extension of a lemma presented in [23] to bound the value of $\|v_i - x^*\|$. This lemma is presented below.

**Lemma A.10 (Simple Lemma on Non-Negative Sequences).**
*Assume that the nonnegative sequences $(u_k)_{k\geq 0}$ and $(a_k)_{k\geq 0}$ satisfy the following recursion for all $k \geq 0$:*

$$u_k^2 \leq S_k + \sum_{i=1}^{k}a_i u_i, \quad (30)$$

*where $(S_k)_{k\geq 0}$ is an increasing sequence such that $S_0 \geq u_0^2$. Then,*

$$u_k \leq \frac{1}{2}\sum_{i=1}^{k}a_i + \sqrt{\left(\frac{1}{2}\sum_{i=1}^{k}a_i\right)^2 + S_k}. \quad (31)$$

*Moreover,*

$$S_k + \sum_{i=1}^{k}a_i u_i \leq \left(\sqrt{S_k} + \sum_{i=1}^{k}a_i\right)^2.$$

*Proof.* The first part—that is, Eq. (31)—is exactly Lemma 1 from [23]. The proof is in their appendix. Then, by calling $b_k$ the right-hand side of (31), we have that for all $k \geq 1$, $u_k \leq b_k$. Furthermore $(b_k)_{k\geq 0}$ is increasing and we have

$$S_k + \sum_{i=1}^{k}a_i u_i \leq S_k + \sum_{i=1}^{k}a_i b_i \leq S_k + \left(\sum_{i=1}^{k}a_i\right)b_k = b_k^2,$$

and using the inequality $\sqrt{x+y} \leq \sqrt{x} + \sqrt{y}$, we have

$$b_k = \frac{1}{2}\sum_{i=1}^{k}a_i + \sqrt{\left(\frac{1}{2}\sum_{i=1}^{k}a_i\right)^2 + S_k} \leq \frac{1}{2}\sum_{i=1}^{k}a_i + \sqrt{\left(\frac{1}{2}\sum_{i=1}^{k}a_i\right)^2} + \sqrt{S_k} = \sqrt{S_k} + \sum_{i=1}^{k}a_i.$$

As a result,

$$S_k + \sum_{i=1}^{k}a_i u_i \leq b_k^2 \leq \left(\sqrt{S_k} + \sum_{i=1}^{k}a_i\right)^2.$$

$\square$

We are now in shape to conclude the proof of Theorem A.3. We apply the previous lemma to (29):

$$\frac{1}{\lambda_k}\left(\frac{\gamma_k}{2}\|x^* - v_k\|^2 + F(x_k) - F^*\right) \le \phi_0(x^*) - F^* + \sum_{i=1}^{k}\frac{\varepsilon_i}{\lambda_i} + \sum_{i=1}^{k}\frac{\sqrt{2\varepsilon_i\gamma_i}}{\lambda_i}\|x^* - v_i\|.$$

Since $F(x_k) - F^* \ge 0$, we have

$$\underbrace{\frac{\gamma_k}{2\lambda_k}\|x^* - v_k\|^2}_{u_k^2} \le \underbrace{\phi_0(x^*) - F^* + \sum_{i=1}^{k}\frac{\varepsilon_i}{\lambda_i}}_{S_k} + \sum_{i=1}^{k}\underbrace{\frac{\sqrt{2\varepsilon_i\gamma_i}}{\lambda_i}\|x^* - v_i\|}_{a_i u_i},$$

with

$$u_i = \sqrt{\frac{\gamma_i}{2\lambda_i}}\|x^* - v_i\| \quad\text{and}\quad a_i = 2\sqrt{\frac{\varepsilon_i}{\lambda_i}} \text{ and } S_k = \phi_0(x^*) - F^* + \sum_{i=1}^{k}\frac{\varepsilon_i}{\lambda_i}.$$

Then by Lemma A.10, we have

$$F(x_k) - F^* \le \lambda_k\left(S_k + \sum_{i=1}^{k}a_i u_i\right) \le \lambda_k\left(\sqrt{S_k} + \sum_{i=1}^{k}a_i\right)^2 = \lambda_k\left(\sqrt{S_k} + 2\sum_{i=1}^{k}\sqrt{\frac{\varepsilon_i}{\lambda_i}}\right)^2,$$

which is the desired result.

# B Proofs of the Main Theorems and Propositions

## B.1 Proof of Theorem 3.1

*Proof.* We simply use Theorem A.3 and specialize it to the choice of parameters. The initialization $\alpha_0 = \sqrt{q}$ leads to a particularly simple form of the algorithm, where $\alpha_k = \sqrt{q}$ for all $k \ge 0$. Therefore, the sequence $(\lambda_k)_{k\ge 0}$ from Theorem A.3 is also simple. For all $k \ge 0$, we indeed have $\lambda_k = (1 - \sqrt{q})^k$. To upper-bound the quantity $S_k$ from Theorem A.3, we now remark that $\gamma_0 = \mu$ and thus, by strong convexity of $F$,

$$F(x_0) + \frac{\gamma_0}{2}\|x_0 - x^*\|^2 - F^* \le 2(F(x_0) - F^*).$$

Therefore,

$$\sqrt{S_k} + 2\sum_{i=1}^{k}\sqrt{\frac{\varepsilon_i}{\lambda_i}} = \sqrt{F(x_0) + \frac{\gamma_0}{2}\|x_0 - x^*\|^2 - F^* + \sum_{i=1}^{k}\frac{\varepsilon_i}{\lambda_i}} + 2\sum_{i=1}^{k}\sqrt{\frac{\varepsilon_i}{\lambda_i}}$$

$$\le \sqrt{F(x_0) + \frac{\gamma_0}{2}\|x_0 - x^*\|^2 - F^*} + 3\sum_{i=1}^{k}\sqrt{\frac{\varepsilon_i}{\lambda_i}}$$

$$\le \sqrt{2(F(x_0) - F^*)} + 3\sum_{i=1}^{k}\sqrt{\frac{\varepsilon_i}{\lambda_i}}$$

$$= \sqrt{2(F(x_0) - F^*)}\left[1 + \sum_{i=1}^{k}\underbrace{\left(\sqrt{\frac{1-\rho}{1-\sqrt{q}}}\right)^i}_{\eta}\right]$$

$$= \sqrt{2(F(x_0) - F^*)}\,\frac{\eta^{k+1} - 1}{\eta - 1}$$

$$\le \sqrt{2(F(x_0) - F^*)}\,\frac{\eta^{k+1}}{\eta - 1}.$$

Therefore, Theorem A.3 combined with the previous inequality gives us

$$F(x_k) - F^* \leq 2\lambda_k(F(x_0) - F^*) \left(\frac{\eta^{k+1}}{\eta - 1}\right)^2$$

$$= 2\left(\frac{\eta}{\eta - 1}\right)^2 (1 - \rho)^k (F(x_0) - F^*)$$

$$= 2\left(\frac{\sqrt{1 - \rho}}{\sqrt{1 - \rho} - \sqrt{1 - \sqrt{q}}}\right)^2 (1 - \rho)^k (F(x_0) - F^*)$$

$$= 2\left(\frac{1}{\sqrt{1 - \rho} - \sqrt{1 - \sqrt{q}}}\right)^2 (1 - \rho)^{k+1} (F(x_0) - F^*).$$

Since $\sqrt{1 - x} + \frac{x}{2}$ is decreasing in $[0, 1]$, we have $\sqrt{1 - \rho} + \frac{\rho}{2} \geq \sqrt{1 - \sqrt{q}} + \frac{\sqrt{q}}{2}$. Consequently,

$$F(x_k) - F^* \leq \frac{8}{(\sqrt{q} - \rho)^2}(1 - \rho)^{k+1}(F(x_0) - F^*).$$

$\square$

## B.2   Proof of Proposition 3.2

To control the number of calls of $\mathcal{M}$, we need to upper bound $G_k(x_{k-1}) - G_k^*$ which is given by the following lemma:

**Lemma B.1 (Relation between $G_k(x_{k-1})$ and $\varepsilon_{k-1}$).**
*Let $(x_k)_{k\geq 0}$ and $(y_k)_{k\geq 0}$ be generated by Algorithm 1. Remember that by definition of $x_{k-1}$,*
$$G_{k-1}(x_{k-1}) - G_{k-1}^* \leq \varepsilon_{k-1}.$$
*Then, we have*

$$G_k(x_{k-1}) - G_k^* \leq 2\varepsilon_{k-1} + \frac{\kappa^2}{\kappa + \mu}\|y_{k-1} - y_{k-2}\|^2. \tag{32}$$

*Proof.* We first remark that for any $x, y$ in $\mathbb{R}^p$, we have
$$G_k(x) - G_{k-1}(x) = G_k(y) - G_{k-1}(y) + \kappa\langle y - x, y_{k-1} - y_{k-2}\rangle, \quad \forall k \geq 2,$$
which can be shown by using the respective definitions of $G_k$ and $G_{k-1}$ and manipulate the quadratic term resulting from the difference $G_k(x) - G_{k-1}(x)$.

Plugging $x = x_{k-1}$ and $y = x_k^*$ in the previous relation yields

$$\begin{aligned}
G_k(x_{k-1}) - G_k^* &= G_{k-1}(x_{k-1}) - G_{k-1}(x_k^*) + \kappa\langle x_k^* - x_{k-1}, y_{k-1} - y_{k-2}\rangle \\
&= G_{k-1}(x_{k-1}) - G_{k-1}^* + G_{k-1}^* - G_{k-1}(x_k^*) + \kappa\langle x_k^* - x_{k-1}, y_{k-1} - y_{k-2}\rangle \\
&\leq \varepsilon_{k-1} + G_{k-1}^* - G_{k-1}(x_k^*) + \kappa\langle x_k^* - x_{k-1}, y_{k-1} - y_{k-2}\rangle \\
&\leq \varepsilon_{k-1} - \frac{\mu + \kappa}{2}\|x_k^* - x_{k-1}^*\|^2 + \kappa\langle x_k^* - x_{k-1}, y_{k-1} - y_{k-2}\rangle,
\end{aligned} \tag{33}$$

where the last inequality comes from the strong convexity inequality of

$$G_{k-1}(x_k^*) \geq G_{k-1}^* + \frac{\mu + \kappa}{2}\|x_k^* - x_{k-1}^*\|^2.$$

Moreover, from the inequality $\langle x, y\rangle \leq \frac{1}{2}\|x\|^2 + \frac{1}{2}\|y\|^2$, we also have

$$\kappa\langle x_k^* - x_{k-1}^*, y_{k-1} - y_{k-2}\rangle \leq \frac{\mu + \kappa}{2}\|x_k^* - x_{k-1}^*\|^2 + \frac{\kappa^2}{2(\kappa + \mu)}\|y_{k-1} - y_{k-2}\|^2, \tag{34}$$

and

$$\kappa\langle x_{k-1}^* - x_{k-1}, y_{k-1} - y_{k-2}\rangle \leq \frac{\mu + \kappa}{2}\|x_{k-1}^* - x_{k-1}\|^2 + \frac{\kappa^2}{2(\kappa + \mu)}\|y_{k-1} - y_{k-2}\|^2$$

$$\leq \varepsilon_{k-1} + \frac{\kappa^2}{2(\kappa + \mu)}\|y_{k-1} - y_{k-2}\|^2. \tag{35}$$

Summing inequalities (33), (34) and (35) gives the desired result. $\square$

Next, we need to upper-bound the term $\|y_{k-1} - y_{k-2}\|^2$, which was also required in the convergence proof of the accelerated SDCA algorithm [26]. We follow here their methodology.

**Lemma B.2 (Control of the term $\|y_{k-1} - y_{k-2}\|^2$.).**
*Let us consider the iterates $(x_k)_{k \geq 0}$ and $(y_k)_{k \geq 0}$ produced by Algorithm 1, and define*

$$\delta_k = C(1 - \rho)^{k+1}(F(x_0) - F^*),$$

*which appears in Theorem 3.1 and which is such that $F(x_k) - F^* \leq \delta_k$. Then, for any $k \geq 3$,*

$$\|y_{k-1} - y_{k-2}\|^2 \leq \frac{72}{\mu} \delta_{k-3}.$$

*Proof.* We follow here [26]. By definition of $y_k$, we have

$$
\begin{aligned}
\|y_{k-1} - y_{k-2}\| &= \|x_{k-1} + \beta_{k-1}(x_{k-1} - x_{k-2}) - x_{k-2} - \beta_{k-2}(x_{k-2} - x_{k-3})\| \\
&\leq (1 + \beta_{k-1})\|x_{k-1} - x_{k-2}\| + \beta_{k-2}\|x_{k-2} - x_{k-3}\| \\
&\leq 3 \max\{\|x_{k-1} - x_{k-2}\|, \|x_{k-2} - x_{k-3}\|\},
\end{aligned}
$$

where $\beta_k$ is defined in (6). The last inequality was due to the fact that $\beta_k \leq 1$. Indeed, the specific choice of $\alpha_0 = \sqrt{q}$ in Theorem A.3 leads to $\beta_k = \frac{\sqrt{q} - q}{\sqrt{q} + q} \leq 1$ for all $k$. Note, however, that this relation $\beta_k \leq 1$ is true regardless of the choice of $\alpha_0$:

$$\beta_k^2 = \frac{(\alpha_{k-1} - \alpha_{k-1}^2)^2}{(\alpha_{k-1}^2 + \alpha_k)^2} = \frac{\alpha_{k-1}^2 + \alpha_{k-1}^4 - 2\alpha_{k-1}^3}{\alpha_k^2 + 2\alpha_k\alpha_{k-1}^2 + \alpha_{k-1}^4} = \frac{\alpha_{k-1}^2 + \alpha_{k-1}^4 - 2\alpha_{k-1}^3}{\alpha_{k-1}^2 + \alpha_{k-1}^4 + q\alpha_k + \alpha_k\alpha_{k-1}^2} \leq 1,$$

where the last equality uses the relation $\alpha_k^2 + \alpha_k\alpha_{k-1}^2 = \alpha_{k-1}^2 + q\alpha_k$ from Algorithm 1. To conclude the lemma, we notice that by triangle inequality

$$\|x_k - x_{k-1}\| \leq \|x_k - x^*\| + \|x_{k-1} - x^*\|,$$

and by strong convexity of $F$

$$\frac{\mu}{2}\|x_k - x^*\|^2 \leq F(x_k) - F(x^*) \leq \delta_k.$$

As a result,

$$
\begin{aligned}
\|y_{k-1} - y_{k-2}\|^2 &\leq 9 \max\{\|x_{k-1} - x_{k-2}\|^2, \|x_{k-2} - x_{k-3}\|^2\} \\
&\leq 36 \max\{\|x_{k-1} - x^*\|^2, \|x_{k-2} - x^*\|^2, \|x_{k-3} - x^*\|^2\} \\
&\leq \frac{72}{\mu} \delta_{k-3}.
\end{aligned}
$$

$\square$

We are now in shape to conclude the proof of Proposition 3.2.

By Proposition B.1 and lemma B.2, we have for all $k \geq 3$,

$$G_k(x_{k-1}) - G_k^* \leq 2\varepsilon_{k-1} + \frac{\kappa^2}{\kappa + \mu} \frac{72}{\mu} \delta_{k-3} \leq 2\varepsilon_{k-1} + \frac{72\kappa}{\mu} \delta_{k-3}.$$

Let $(z_t)_{t \geq 0}$ be the sequence of using $\mathcal{M}$ to solve $G_k$ with initialization $z_0 = x_{k-1}$. By assumption (8), we have

$$G_k(z_t) - G_k^* \leq A(1 - \tau_{\mathcal{M}})^t(G_k(x_{k-1}) - G_k^*) \leq A e^{-\tau_{\mathcal{M}} t}(G_k(x_{k-1}) - G_k^*).$$

The number of iterations $T_{\mathcal{M}}$ of $\mathcal{M}$ to guarantee an accuracy of $\varepsilon_k$ needs to satisfy

$$A e^{-\tau_{\mathcal{M}} T_{\mathcal{M}}}(G_k(x_{k-1}) - G_k^*) \leq \varepsilon_k,$$

which gives

$$T_{\mathcal{M}} = \left\lceil \frac{1}{\tau_{\mathcal{M}}} \log\left(\frac{A(G_k(x_{k-1}) - G_k^*)}{\varepsilon_k}\right) \right\rceil. \tag{36}$$

Then, it remains to upper-bound

$$\frac{G_k(x_{k-1}) - G_k^*}{\varepsilon_k} \leq \frac{2\varepsilon_{k-1} + \frac{72\kappa}{\mu}\delta_{k-3}}{\varepsilon_k} = \frac{2(1-\rho) + \frac{72\kappa}{\mu} \cdot \frac{9C}{2}}{(1-\rho)^2} = \frac{2}{1-\rho} + \frac{2592\kappa}{\mu(1-\rho)^2(\sqrt{q}-\rho)^2}.$$

Let us denote $R$ the right-hand side. We remark that this upper bound holds for $k \geq 3$. We now consider the cases $k = 1$ and $k = 2$.

When $k = 1$, $G_1(x) = F(x) + \frac{\kappa}{2}\|x - y_0\|^2$. Note that $x_0 = y_0$, then $G_1(x_0) = F(x_0)$. As a result,

$$G_1(x_0) - G_1^* = F(x_0) - F(x_1^*) - \frac{\kappa}{2}\|x_1^* - y_0\|^2 \leq F(x_0) - F(x_1^*) \leq F(x_0) - F^*.$$

Therefore,

$$\frac{G_1(x_0) - G_1^*}{\varepsilon_1} \leq \frac{F(x_0) - F^*}{\varepsilon_1} = \frac{9}{2(1-\rho)} \leq R.$$

When $k = 2$, we remark that $y_1 - y_0 = (1 + \beta_1)(x_1 - x_0)$. Then, by following similar steps as in the proof of Lemma B.2, we have

$$\|y_1 - y_0\|^2 \leq 4\|x_1 - x_0\|^2 \leq \frac{32\delta_0}{\mu},$$

which is smaller than $\frac{72\delta_{-1}}{\mu}$. Therefore, the previous steps from the case $k \geq 3$ apply and $\frac{G_2(x_1) - G_2^*}{\varepsilon_2} \leq R$. Thus, for any $k \geq 1$,

$$T_{\mathcal{M}} \leq \left\lceil \frac{\log(AR)}{\tau_{\mathcal{M}}} \right\rceil, \tag{37}$$

which concludes the proof.

## B.3 Proof of Theorem 3.3.

We will again Theorem A.3 and specialize it to the choice of parameters. To apply it, the following Lemma will be useful to control the growth of $(\lambda_k)_{k \geq 0}$.

**Lemma B.3** (Growth of the Sequence $(\lambda_k)_{k \geq 0}$).
*Let $(\lambda_k)_{k \geq 0}$ be the sequence defined in (15) where $(\alpha_k)_{k \geq 0}$ is produced by Algorithm 1 with $\alpha_0 = \frac{\sqrt{5}-1}{2}$ and $\mu = 0$. Then, we have the following bounds for all $k \geq 0$,*

$$\frac{4}{(k+2)^2} \geq \lambda_k \geq \frac{2}{(k+2)^2}.$$

*Proof.* Note that by definition of $\alpha_k$, we have for all $k \geq 1$,

$$\alpha_k^2 = (1 - \alpha_k)\alpha_{k-1}^2 = \prod_{i=1}^{k}(1 - \alpha_i)\alpha_0^2 = \lambda_{k+1}\frac{\alpha_0^2}{1 - \alpha_0} = \lambda_{k+1}.$$

With the choice of $\alpha_0$, the quantity $\gamma_0$ defined in (17) is equal to $\kappa$. By Lemma A.4, we have $\lambda_k \leq \frac{4}{(k+2)^2}$ for all $k \geq 0$ and thus $\alpha_k \leq \frac{2}{k+3}$ for all $k \geq 1$ (it is also easy to check numerically that this is also true for $k = 0$ since $\frac{\sqrt{5}-1}{2} \approx 0.62 \leq \frac{2}{3}$). We now have all we need to conclude the lemma:

$$\lambda_k = \prod_{i=0}^{k-1}(1 - \alpha_i) \geq \prod_{i=0}^{k-1}\left(1 - \frac{2}{i+3}\right) = \frac{2}{(k+2)(k+1)} \geq \frac{2}{(k+2)^2}.$$

$\square$

With this lemma in hand, we may now proceed and apply Theorem A.3. We have remarked in the proof of the previous lemma that $\gamma_0 = \kappa$. Then,

$$\sqrt{S_k} + 2\sum_{i=1}^{k}\sqrt{\frac{\varepsilon_i}{\lambda_i}} = \sqrt{F(x_0) - F^* + \frac{\kappa}{2}\|x_0 - x^*\|^2 + \sum_{i=1}^{k}\frac{\varepsilon_i}{\lambda_i}} + 2\sum_{i=1}^{k}\sqrt{\frac{\varepsilon_i}{\lambda_i}}$$

$$\leq \sqrt{F(x_0) - F^* + \frac{\kappa}{2}\|x_0 - x^*\|^2} + 3\sum_{i=1}^{k}\sqrt{\frac{\varepsilon_i}{\lambda_i}}$$

$$\leq \sqrt{\frac{\kappa}{2}\|x_0 - x^*\|^2} + \sqrt{F(x_0) - F^*}\left(1 + \sum_{i=1}^{k}\frac{1}{(i+2)^{1+\eta/2}}\right),$$

where the last inequality uses Lemma B.3 to upper-bound the ratio $\varepsilon_i/\lambda_i$. Moreover,

$$\sum_{i=1}^{k}\frac{1}{(i+2)^{1+\eta/2}} \leq \sum_{i=2}^{\infty}\frac{1}{i^{1+\eta/2}} \leq \int_{1}^{\infty}\frac{1}{x^{1+\eta/2}}\,\mathrm{d}x = \frac{2}{\eta}.$$

Therefore, by (16) from Theorem A.3,

$$F(x_k) - F^* \leq \lambda_k \left(\sqrt{S_k} + 2\sum_{i=1}^{k}\sqrt{\frac{\varepsilon_i}{\lambda_i}}\right)^2$$

$$\leq \frac{4}{(k+2)^2}\left(\sqrt{F(x_0) - F^*}\left(1 + \frac{2}{\eta}\right) + \sqrt{\frac{\kappa}{2}\|x_0 - x^*\|^2}\right)^2$$

$$\leq \frac{8}{(k+2)^2}\left(\left(1 + \frac{2}{\eta}\right)^2(F(x_0) - F^*) + \frac{\kappa}{2}\|x_0 - x^*\|^2\right).$$

The last inequality uses $(a + b)^2 \leq 2(a^2 + b^2)$.

## B.4   Proof of Proposition 3.4

When $\mu = 0$, we remark that Proposition B.1 still holds but Lemma B.2 does not. The main difficulty is thus to find another way to control the quantity $\|y_{k-1} - y_{k-2}\|$.

Since $F(x_k) - F^*$ is bounded by Theorem 3.3, we may use the bounded level set assumptions to ensure that there exists $B > 0$ such that $\|x_k - x^*\| \leq B$ for any $k \geq 0$ where $x^*$ is a minimizer of $F$. We can now follow similar steps as in the proof of Lemma B.2, and show that

$$\|y_{k-1} - y_{k-2}\|^2 \leq 36B^2.$$

Then by Proposition B.1,

$$G_k(x_{k-1}) - G_k^* \leq 2\varepsilon_{k-1} + 36\kappa B^2.$$

Since $\kappa > 0$, $G_k$ is strongly convex, then using the same argument as in the strongly convex case, the number of calls for $\mathcal{M}$ is given by

$$\left\lceil \frac{1}{\tau_{\mathcal{M}}}\log\left(\frac{A(G_k(x_{k-1}) - G_k^*)}{\varepsilon_k}\right)\right\rceil. \tag{38}$$

Again, we need to upper bound it

$$\frac{G_k(x_{k-1}) - G_k^*}{\varepsilon_k} \leq \frac{2\varepsilon_{k-1} + 36\kappa B^2}{\varepsilon_k} = \frac{2(k+1)^{4+\eta}}{(k+2)^{4+\eta}} + \frac{162\kappa B^2(k+2)^{4+\eta}}{(F(x_0) - F^*)}.$$

The right hand side is upper-bounded by $O((k+2)^{4+\eta})$. Plugging this relation into (38) gives the desired result.

# C Derivation of Global Convergence Rates

We give here a generic "template" for computing the *optimal choice of* $\kappa$ to accelerate a given algorithm $\mathcal{M}$, and therefore compute the rate of convergence of the accelerated algorithm $\mathcal{A}$.

We assume here that $\mathcal{M}$ is a *randomized* first-order optimization algorithm, *i.e.* the iterates $(x_k)$ generated by $\mathcal{M}$ are a sequence of random variables; specialization to a deterministic algorithm is straightforward. Also, for the sake of simplicity, we shall use simple notations to denote the stopping time to reach accuracy $\varepsilon$. Definition and notation using filtrations, $\sigma$-algebras, etc. are unnecessary for our purpose here where the quantity of interest has a clear interpretation.

Assume that algorithm $\mathcal{M}$ enjoys a linear rate of convergence, in expectation. There exists constants $\mathcal{C}_{\mathcal{M},F}$ and $\tau_{\mathcal{M},F}$ such that the sequence of iterates $(x_k)_{k \geq 0}$ for minimizing a strongly-convex objective $F$ satisfies

$$\mathbb{E}\left[F(x_k) - F^*\right] \leq \mathcal{C}_{\mathcal{M},F}\left(1 - \tau_{\mathcal{M},F}\right)^k. \tag{39}$$

Define the random variable $T_{\mathcal{M},F}(\varepsilon)$ (stopping time) corresponding to the minimum number of iterations to guarantee an accuracy $\varepsilon$ in the course of running $\mathcal{M}$

$$T_{\mathcal{M},F}(\varepsilon) := \inf\{k \geq 1, \ F(x_k) - F^* \leq \varepsilon\} \tag{40}$$

Then, an upper bound on the expectation is provided by the following lemma.

**Lemma C.1 (Upper Bound on the expectation of $T_{\mathcal{M},F}(\varepsilon)$).**
*Let $\mathcal{M}$ be an optimization method with the expected rate of convergence (39). Then,*

$$\mathbb{E}[T_{\mathcal{M}}(\varepsilon)] \leq \frac{1}{\tau_{\mathcal{M}}}\log\left(\frac{2C_{\mathcal{M}}}{\tau_{\mathcal{M}} \cdot \varepsilon}\right) + 1 = \tilde{O}\left(\frac{1}{\tau_{\mathcal{M}}}\log\left(\frac{C_{\mathcal{M}}}{\varepsilon}\right)\right), \tag{41}$$

*where we have dropped the dependency in $F$ to simplify the notation.*

*Proof.* We abbreviate $\tau_{\mathcal{M}}$ by $\tau$. Set

$$T_0 = \frac{1}{\tau}\log\left(\frac{1}{1 - e^{-\tau}}\frac{C_{\mathcal{M}}}{\varepsilon}\right).$$

For any $k \geq 0$, we have

$$\mathbb{E}[F(x_k) - F^*] \leq C_{\mathcal{M}}(1 - \tau)^k \leq C_{\mathcal{M}}\,e^{-k\tau}.$$

By Markov's inequality,

$$\mathbb{P}[F(x_k) - F^* > \varepsilon] = \mathbb{P}[T_{\mathcal{M}}(\varepsilon) > k] \leq \frac{\mathbb{E}[F(x_k) - F^*]}{\varepsilon} \leq \frac{C_{\mathcal{M}}\,e^{-k\tau}}{\varepsilon}. \tag{42}$$

Together with the fact $\mathbb{P} \leq 1$ and $k \geq 0$. We have

$$\mathbb{P}[T_{\mathcal{M}}(\varepsilon) \geq k + 1] \leq \min\left\{\frac{C_{\mathcal{M}}}{\varepsilon}e^{-k\tau}, 1\right\}.$$

Therefore,

$$\mathbb{E}[T_{\mathcal{M}}(\varepsilon)] = \sum_{k=1}^{\infty}\mathbb{P}[T_{\mathcal{M}}(\varepsilon) \geq k] = \sum_{k=1}^{T_0}\mathbb{P}[T_{\mathcal{M}}(\varepsilon) \geq k] + \sum_{k=T_0+1}^{\infty}\mathbb{P}[T_{\mathcal{M}}(\varepsilon) \geq k]$$

$$\leq T_0 + \sum_{k=T_0}^{\infty}\frac{C_{\mathcal{M}}}{\varepsilon}e^{-k\tau} = T_0 + \frac{C_{\mathcal{M}}}{\varepsilon}e^{-T_0\tau}\sum_{k=0}^{\infty}e^{-k\tau}$$

$$= T_0 + \frac{C_{\mathcal{M}}}{\varepsilon}\frac{e^{-\tau T_0}}{1 - e^{-\tau}} = T_0 + 1.$$

As simple calculation shows that for any $\tau \in (0,1)$, $\frac{\tau}{2} \leq 1 - e^{-\tau}$ and then

$$\mathbb{E}[T_{\mathcal{M}}(\varepsilon)] \leq T_0 + 1 = \frac{1}{\tau}\log\left(\frac{1}{1 - e^{-\tau}}\frac{C_{\mathcal{M}}}{\varepsilon}\right) + 1 \leq \frac{1}{\tau}\log\left(\frac{2C_{\mathcal{M}}}{\tau\varepsilon}\right) + 1.$$

$$\square$$

Note that the previous lemma mirrors Eq. (36-37) in the proof of Prop. 3.1 in Appendix B. For all optimization methods of interest, the rate $\tau_{\mathcal{M},G_k}$ is independent of $k$ and varies with the parameter $\kappa$. We may now compute the iteration-complexity (in expectation) of the accelerated algorithm $\mathcal{A}$—that is, for a given $\varepsilon$, the expected total number of iterations performed by the method $\mathcal{M}$. Let us now fix $\varepsilon > 0$. Calculating the iteration-complexity decomposes into three steps:

1. Find $\kappa$ that maximizes the ratio $\tau_{\mathcal{M},G_k}/\sqrt{\mu + \kappa}$ for algorithm $\mathcal{M}$ when $F$ is $\mu$-strongly convex. In the non-strongly convex case, we suggest maximizing instead the ratio $\tau_{\mathcal{M},G_k}/\sqrt{L + \kappa}$. Note that the choice of $\kappa$ is less critical for non-strongly convex problems since it only affects multiplicative constants in the global convergence rate.

2. Compute the upper-bound of the number of outer iterations $k_{\text{out}}$ using Theorem 3.1 (for the strongly convex case), or Theorem 3.3 (for the non-strongly convex case), by replacing $\kappa$ by the optimal value found in step 1.

3. Compute the upper-bound of the expected number of inner iterations
$$\max_{k=1,\ldots,k_{\text{out}}} \mathbb{E}[T_{\mathcal{M},G_k}(\varepsilon_k)] \leq k_{\text{in}},$$
by replacing the appropriate quantities in Eq. 41 for algorithm $\mathcal{M}$; for that purpose, the proofs of Propositions 3.2 of 3.4 my be used to upper-bound the ratio $\mathcal{C}_{\mathcal{M},G_k}/\varepsilon_k$, or another dedicated analysis for $\mathcal{M}$ may be required if the constant $\mathcal{C}_{\mathcal{M},G_k}$ does not have the required form $A(G_k(z_0) - G_k^*)$ in (8).

Then, the iteration-complexity (in expectation) denoted Comp. is given by
$$\text{Comp} \leq k_{\text{in}} \times k_{\text{out}} . \tag{43}$$

# D  A Proximal MISO/Finito Algorithm

In this section, we present the algorithm MISO/Finito, and show how to extend it in two ways. First, we propose a proximal version to deal with composite optimization problems, and we analyze its rate of convergence. Second, we show how to remove a large sample condition $n \geq 2L/\mu$, which was necessary for the convergence of the algorithm. The resulting algorithm is a variant of proximal SDCA [25] with a different stepsize and a stopping criterion that does not use duality.

## D.1  The Original Algorithm MISO/Finito

MISO/Finito was proposed in [14] and [7] for solving the following smooth unconstrained convex minimization problem
$$\min_{x \in \mathbb{R}^p} \left\{ f(x) \triangleq \frac{1}{n} \sum_{i=1}^{n} f_i(x) \right\}, \tag{44}$$
where each $f_i$ is differentiable with $L$-Lipschitz continuous derivatives and $\mu$-strongly convex. At iteration $k$, the algorithm updates a list of lower bounds $d_i^k$ of the functions $f_i$, by randomly picking up one index $i_k$ among $\{1, \cdots, n\}$ and performing the following update
$$d_i^k(x) = \begin{cases} f_i(x_{k-1}) + \langle \nabla f_i(x_{k-1}), x - x_{k-1} \rangle + \frac{\mu}{2}\|x - x_{k-1}\|^2 & \text{if } i = i_k \\ d_i^{k-1}(x) & \text{otherwise} \end{cases},$$
which is a lower bound of $f_i$ because of the $\mu$-strong convexity of $f_i$. Equivalently, one may perform the following updates
$$z_i^k = \begin{cases} x_{k-1} - \frac{1}{\mu}\nabla f_i(x_{k-1}) & \text{if } i = i_k \\ z_i^{k-1} & \text{otherwise} \end{cases},$$
and all functions $d_i^k$ have the form
$$d_i^k(x) = c_i^k + \frac{\mu}{2}\|x - z_i^k\|^2,$$
where $c_i^k$ is a constant. Then, MISO/Finito performs the following minimization to produce the iterate $(x_k)$:
$$x_k = \arg\min_{x \in \mathbb{R}^p} \frac{1}{n} \sum_{i=1}^{n} d_i^k(x) = \frac{1}{n} \sum_{i=1}^{n} z_i^k,$$

which is equivalent to

$$x_k \leftarrow x_{k-1} - \frac{1}{n} \left( z_{i_k}^k - z_{i_k}^{k-1} \right).$$

In many machine learning problems, it is worth remarking that each function $f_i(x)$ has the specific form $f_i(x) = l_i(\langle x, w_i \rangle) + \frac{\mu}{2}\|x\|^2$. In such cases, the vectors $z_i^k$ can be obtained by storing only $O(n)$ scalars.[3] The main convergence result of [14] is that the procedure above converges with a linear rate of convergence of the form (3), with $\tau_{\text{MISO}} = 1/3n$ (also refined in $1/2n$ in [7]), when the large sample size constraint $n \geq 2L/\mu$ is satisfied.

Removing this condition and extending MISO to the composite optimization problem (1) is the purpose of the next section.

### D.2 Proximal MISO

We now consider the composite optimization problem below,

$$\min_{x \in \mathbb{R}^p} \left\{ F(x) = \frac{1}{n} \sum_{i=1}^{n} f_i(x) + \psi(x) \right\},$$

where the functions $f_i$ are differentiable with $L$-Lipschitz derivatives and $\mu$-strongly convex. As in typical composite optimization problems, $\psi$ is convex but not necessarily differentiable. We assume that the proximal operator of $\psi$ can be computed easily. The algorithm needs to be initialized with some lower bounds for the functions $f_i$:

$$f_i(x) \geq \frac{\mu}{2}\|x - z_i^0\|^2 + c_i^0, \tag{A1}$$

which are guaranteed to exist due to the $\mu$-strong convexity of $f_i$. For typical machine learning applications, such initialization is easy. For example, logistic regression with $\ell_2$-regularization satisfies (A1) with $z_i^0 = 0$ and $c_i^0 = 0$. Then, the MISO-Prox scheme is given in Algorithm 2. Note that if no simple initialization is available, we may consider any initial estimate $\bar{z}_0$ in $\mathbb{R}^p$ and define $z_i^0 = \bar{z}_0 - (1/\mu)\nabla f_i(\bar{z}_0)$, which requires performing one pass over the data.

Then, we remark that under the large sample size condition $n \geq 2L/\mu$, we have $\delta = 1$ and the update of the quantities $z_i^k$ in (45) is the same as in the original MISO/Finito algorithm. As we will see in the convergence analysis, the choice of $\delta$ ensures convergence of the algorithm even in the small sample size regime $n < 2L/\mu$.

**Relation with Proximal SDCA [25].**   The algorithm MISO-Prox is almost identical to variant 5 of proximal SDCA [25], which performs the same updates with $\delta = \mu n/(L + \mu n)$ instead of $\delta = \min(1, \frac{\mu n}{2(L-\mu)})$. It is however not clear that MISO-Prox actually performs dual ascent steps in the sense of SDCA since the proof of convergence of SDCA cannot be directly modified to use the stepsize of proximal MISO and furthermore, the convergence proof of MISO-Prox does not use the concept of duality. Another difference lies in the optimality certificate of the algorithms. Whereas Proximal-SDCA provides a certificate in terms of linear convergence of a duality gap based on Fenchel duality, Proximal-SDCA ensures linear convergence of a gap that relies on strong convexity but not on the Fenchel dual (at least explicitly).

**Optimality Certificate and Stopping Criterion.**   Similar to the original MISO algorithm, Proximal MISO maintains a list $(d_i^k)$ of lower bounds of the functions $f_i$, which are updated in the following fashion

$$d_i^k(x) = \begin{cases} (1-\delta)d_i^{k-1}(x) + \delta \left( f_i(x_{k-1}) + \langle \nabla f_i(x_{k-1}), x - x_{k-1} \rangle + \frac{\mu}{2}\|x - x_{k-1}\|^2 \right) & \text{if } i = i_k \\ d_i^{k-1}(x) & \text{otherwise} \end{cases} \tag{46}$$

**Algorithm 2** MISO-Prox: an improved MISO algorithm with proximal support.

**input** $(z_i^0)_{i=1,\ldots,n}$ such that (A1) holds; $N$ (number of iterations);
1: initialize $\bar{z}_0 = \frac{1}{n} \sum_{i=1}^{n} z_i^0$ and $x_0 = \text{prox}_{\psi/\mu}[\bar{z}_0]$;
2: define $\delta = \min\left(1, \frac{\mu n}{2(L-\mu)}\right)$;
3: **for** $k = 1, \ldots, N$ **do**
4:   randomly pick up an index $i_k$ in $\{1, \ldots, n\}$;
5:   update

$$
z_i^k = \begin{cases} (1-\delta)z_i^{k-1} + \delta\left(x_{k-1} - \frac{1}{\mu}\nabla f_i(x_{k-1})\right) & \text{if } i = i_k \\ z_i^{k-1} & \text{otherwise} \end{cases}
$$

$$
\bar{z}_k = \bar{z}_{k-1} - \frac{1}{n}\left(z_{i_k}^k - z_{i_k}^{k-1}\right) = \frac{1}{n}\sum_{i=1}^{n} z_i^k \tag{45}
$$

$$
x_k = \text{prox}_{\psi/\mu}[\bar{z}_k].
$$

6: **end for**
**output** $x_N$ (final estimate).

Then, the following function is a lower bound of the objective $F$:

$$
D_k(x) = \frac{1}{n}\sum_{i=1}^{n} d_i^k(x) + \psi(x), \tag{47}
$$

and the update (45) can be shown to exactly minimize $D_k$. As a lower bound of $F$, we have that $D_k(x_k) \leq F^*$ and thus

$$
F(x_k) - F^* \leq F(x_k) - D_k(x_k).
$$

The quantity $F(x_k) - D_k(x_k)$ can then be interpreted as an optimality gap, and the analysis below will show that it converges linearly to zero. In practice, it also provides a convenient stopping criterion, which yields Algorithm 3.

**Algorithm 3** MISO-Prox with stopping criterion.

**input** $(z_i^0, c_i^0)_{i=1,\ldots,n}$ such that (A1) holds; $\varepsilon$ (target accuracy);
1: initialize $\bar{z}_0 = \frac{1}{n}\sum_{i=1}^{n} z_i^0$ and $c_i'^0 = c_i^0 + \frac{\mu}{2}\|\bar{z}_0\|^2$ for all $i$ in $\{1, \ldots, n\}$ and $x_0 = \text{prox}_{\psi/\mu}[\bar{z}_0]$;
2: Define $\delta = \min\left(1, \frac{\mu n}{2(L-\mu)}\right)$ and $k = 0$;
3: **while** $\frac{1}{n}\sum_{i=1}^{n} f_i(x_k) - c_i'^k + \mu\langle\bar{z}_k, x_k\rangle - \frac{\mu}{2}\|x_k\|^2 > \varepsilon$ **do**
4:   **for** $l = 1, \ldots, n$ **do**
5:     $k \leftarrow k + 1$;
6:     randomly pick up an index $i_k$ in $\{1, \ldots, n\}$;
7:     perform the update (45);
8:     update

$$
c_i'^k = \begin{cases} (1-\delta)c_i'^{k-1} + \delta\left(f_i(x_{k-1}) - \langle\nabla f_i(x_{k-1}), x_{k-1}\rangle + \frac{\mu}{2}\|x_{k-1}\|^2\right) & \text{if } i = i_k \\ c_i'^{k-1} & \text{otherwise} \end{cases}.
$$
$$
\tag{48}
$$

9:   **end for**
10: **end while**
**output** $x_N$ (final estimate such that $F(x_N) - F^* \leq \varepsilon$).

To explain the stopping criterion in Algorithm 3, we remark that the functions $d_i^k$ are quadratic and can be written

$$
d_i^k(x) = c_i^k + \frac{\mu}{2}\|x - z_i^k\|^2 = c_i'^k - \mu\langle x, z_i^k\rangle + \frac{\mu}{2}\|x\|^2, \tag{49}
$$

where the $c_i^k$'s are some constants and $c_i'^k = c_i^k + \frac{\mu}{2}\|z_i^k\|^2$. Equation (48) shows how to update recursively these constants $c_i'^k$, and finally

$$D_k(x_k) = \left(\frac{1}{n}\sum_{i=1}^{n} c_i'^k\right) - \mu\langle x_k, \bar{z}_k\rangle + \frac{\mu}{2}\|x_k\|^2 + \psi(x_k),$$

and

$$F(x_k) - D_k(x_k) = \left(\frac{1}{n}\sum_{i=1}^{n} f_i(x_k) - c_i'^k\right) + \mu\langle x_k, \bar{z}_k\rangle - \frac{\mu}{2}\|x_k\|^2,$$

which justifies the stopping criterion. Since computing $F(x_k)$ requires scanning all the data points, the criterion is only computed every $n$ iterations.

**Convergence Analysis.** The convergence of MISO-Prox is guaranteed by Theorem 4.1 from the main part of paper. Before we prove this theorem, we note that this rate is slightly better than the one proven in MISO [14], which converges as $(1 - \frac{1}{3n})^k$. We start by recalling a classical lemma that provides useful inequalities. Its proof may be found in [17].

**Lemma D.1** (**Classical Quadratic Upper and Lower Bounds**).
*For any function $g : \mathbb{R}^p \to \mathbb{R}$ which is $\mu$-strongly convex and differentiable with $L$-Lipschitz derivatives, we have for all $x, y$ in $\mathbb{R}^p$,*

$$\frac{\mu}{2}\|x - y\|^2 \leq g(x) - g(y) + \langle \nabla g(y), x - y\rangle \leq \frac{L}{2}\|x - y\|^2.$$

To start the proof, we need a sequence of upper and lower bounds involving the functions $D_k$ and $D_{k-1}$. The first one is given in the next lemma

**Lemma D.2** (**Lower Bound on $D_k$**).
*For all $k \geq 1$ and $x$ in $\mathbb{R}^p$,*

$$D_k(x) \geq D_{k-1}(x) - \frac{\delta(L-\mu)}{2n}\|x - x_{k-1}\|^2, \quad \forall x \in \mathbb{R}^p. \tag{50}$$

*Proof.* For any $i \in \{1, \ldots, n\}$, $f_i$ satisfies the assumptions of Lemma D.1, and we have for all $k \geq 0$, $x$ in $\mathbb{R}^p$, and for $i = i_k$,

$$d_i^k(x) = (1 - \delta)d_i^{k-1}(x) + \delta[f_i(x_{k-1}) + \langle \nabla f_i(x_{k-1}), x - x_{k-1}\rangle + \frac{\mu}{2}\|x - x_{k-1}\|^2]$$

$$\geq (1 - \delta)d_i^{k-1}(x) + \delta f_i(x) - \frac{\delta(L-\mu)}{2}\|x - x_{k-1}\|^2$$

$$\geq d_i^{k-1}(x) - \frac{\delta(L-\mu)}{2}\|x - x_{k-1}\|^2,$$

where the definition of $d_i^k$ is given in (46). The first inequality uses Lemma D.1, and the last one uses the inequality $f_i \geq d_i^{k-1}$. From this inequality, we can obtain (50) by simply using $D_k(x) = \sum_{i=1}^{n} d_i^k(x) + \psi(x) = D_{k-1}(x) + \frac{1}{n}\left(d_{i_k}^k(x) - d_{i_k}^{k-1}(x)\right)$. $\qquad\square$

Next, we prove the following lemma to compare $D_k$ and $D_{k-1}$.

**Lemma D.3** (**Relation between $D_k$ and $D_{k-1}$**).
*For all $k \geq 0$, for all $x$ and $y$ in $\mathbb{R}^p$,*

$$D_k(x) - D_k(y) = D_{k-1}(x) - D_{k-1}(y) - \mu\langle \bar{z}_k - \bar{z}_{k-1}, x - y\rangle. \tag{51}$$

*Proof.* Remember that the functions $d_i^k$ are quadratic and have the form (49), that $D_k$ is defined in (47), and that $\bar{z}_k$ minimizes $\frac{1}{n}\sum_{i=1}^{n} d_i^k$. Then, there exists a constant $A_k$ such that

$$D_k(x) = A_k + \frac{\mu}{2}\|x - \bar{z}_k\|^2 + \psi(x).$$

This gives

$$D_k(x) - D_k(y) = \frac{\mu}{2}\|x - \bar{z}_k\|^2 - \frac{\mu}{2}\|y - \bar{z}_k\|^2 + \psi(x) - \psi(y). \tag{52}$$

Similarly,

$$D_{k-1}(x) - D_{k-1}(y) = \frac{\mu}{2}\|x - \bar{z}_{k-1}\|^2 - \frac{\mu}{2}\|y - \bar{z}_{k-1}\|^2 + \psi(x) - \psi(y). \tag{53}$$

Subtracting (52) and (53) gives (51). □

Then, we are able to control the value of $D_k(x_{k-1})$ in the next lemma.

**Lemma D.4** (**Controlling the value** $D_k(x_{k-1})$)**.**
*For any $k \geq 1$,*

$$D_k(x_{k-1}) - D_k(x_k) \leq \frac{\mu}{2}\|\bar{z}_k - \bar{z}_{k-1}\|^2. \tag{54}$$

*Proof.* Using Lemma D.3 with $x = x_{k-1}$ and $y = x_k$ yields

$$D_k(x_{k-1}) - D_k(x_k) = D_{k-1}(x_{k-1}) - D_{k-1}(x_k) - \mu\langle \bar{z}_k - \bar{z}_{k-1}, x_{k-1} - x_k \rangle.$$

Moreover $x_{k-1}$ is the minimum of $D_{k-1}$ which is $\mu$-strongly convex. Thus,

$$D_{k-1}(x_{k-1}) + \frac{\mu}{2}\|x_k - x_{k-1}\|^2 \leq D_{k-1}(x_k).$$

Adding the two previous inequalities gives the first inequality below

$$D_k(x_{k-1}) - D_k(x_k) \leq -\frac{\mu}{2}\|x_k - x_{k-1}\|^2 - \mu\langle \bar{z}_k - \bar{z}_{k-1}, x_{k-1} - x_k \rangle \leq \frac{\mu}{2}\|\bar{z}_k - \bar{z}_{k-1}\|^2,$$

and the last one comes from the basic inequality $\frac{1}{2}\|a\|^2 + \langle a, b\rangle + \frac{1}{2}\|b\|^2 \geq 0$. □

We have now all the inequalities in hand to prove Theorem 4.1.

*Proof of Theorem 4.1.*
We start by giving a lower bound of $D_k(x_{k-1}) - D_{k-1}(x_{k-1})$.
Take $x = x_{k-1}$ in (51). Then, for all $y$ in $\mathbb{R}^p$,

$$D_k(x_{k-1}) - D_{k-1}(x_{k-1}) = D_k(y) - D_{k-1}(y) + \mu\langle \bar{z}_k - \bar{z}_{k-1}, y - x_{k-1} \rangle$$

$$by \ (50) \geq -\frac{\delta(L-\mu)}{2n}\|y - x_{k-1}\|^2 + \mu\langle \bar{z}_k - \bar{z}_{k-1}, y - x_{k-1} \rangle$$

Choose $y$ that maximizes the above quadratic function, i.e.

$$y = x_{k-1} + \frac{n\mu}{\delta(L-\mu)}(\bar{z}_k - \bar{z}_{k-1}),$$

and then

$$D_k(x_{k-1}) - D_{k-1}(x_{k-1}) \geq \frac{n\mu^2}{2\delta(L-\mu)}\|\bar{z}_k - \bar{z}_{k-1}\|^2$$
$$by \ (54) \geq \frac{n\mu}{\delta(L-\mu)}\left[D_k(x_{k-1}) - D_k(x_k)\right]. \tag{55}$$

Then, we start introducing expected values.
By construction

$$D_k(x_{k-1}) = D_{k-1}(x_{k-1}) + \frac{\delta}{n}(f_{i_k}(x_{k-1}) - d_{i_k}^{k-1}(x_{k-1})).$$

After taking expectation, we obtain the relation

$$\mathbb{E}[D_k(x_{k-1})] = \left(1 - \frac{\delta}{n}\right)\mathbb{E}[D_{k-1}(x_{k-1})] + \frac{\delta}{n}\mathbb{E}[F(x_{k-1})]. \tag{56}$$

We now introduce an important quantity

$$\tau = \left(1 - \frac{\delta(L-\mu)}{n\mu}\right)\frac{\delta}{n},$$

and combine (55) with (56) to obtain

$$\tau \mathbb{E}[F(x_{k-1})] - \mathbb{E}[D_k(x_k)] \leq -(1-\tau)\mathbb{E}[D_{k-1}(x_{k-1})].$$

We reformulate this relation as

$$\tau\left(\mathbb{E}[F(x_{k-1})] - F^*\right) + \left(F^* - \mathbb{E}[D_k(x_k)]\right) \leq (1-\tau)\left(F^* - \mathbb{E}[D_{k-1}(x_{k-1})]\right). \qquad (57)$$

On the one hand, since $F(x_{k-1}) \geq F^*$, we have

$$F^* - \mathbb{E}[D_k(x_k)] \leq (1-\tau)\left(F^* - \mathbb{E}[D_{k-1}(x_{k-1})]\right).$$

This is true for any $k \geq 1$, as a result

$$F^* - \mathbb{E}[D_k(x_k)] \leq (1-\tau)^k\left(F^* - D_0(x_0)\right). \qquad (58)$$

On the other hand, since $F^* \geq D_k(x_k)$, then

$$\tau\left(\mathbb{E}[F(x_{k-1})] - F^*\right) \leq (1-\tau)\left(F^* - \mathbb{E}[D_{k-1}(x_{k-1})]\right) \leq (1-\tau)^k\left(F^* - D_0(x_0)\right),$$

which gives us the relation (14) of the theorem. We conclude giving the choice of $\delta$. We choose it to maximize the rate of convergence, which turns to maximize $\tau$. This is a quadratic function, which is maximized at $\delta = \frac{n\mu}{2(L-\mu)}$. However, by definition $\delta \leq 1$. Therefore, the optimal choice of $\delta$ is given by

$$\delta = \min\left\{1, \frac{n\mu}{2(L-\mu)}\right\}.$$

Note now that

1. When $\frac{n\mu}{2(L-\mu)} \leq 1$, we have $\delta = \frac{n\mu}{2(L-\mu)}$ and $\tau = \frac{\mu}{4(L-\mu)}$.

2. When $1 \leq \frac{n\mu}{2(L-\mu)}$, we have $\delta = 1$ and $\tau = \frac{1}{n} - \frac{L-\mu}{n^2\mu} \geq \frac{1}{2n}$.

Therefore, $\tau \geq \min\left(\frac{1}{2n}, \frac{\mu}{4(L-\mu)}\right)$, which concludes the first part of the theorem.

To prove the second part, we use (58) and (14), which gives

$$\begin{aligned}
\mathbb{E}[F(x_k) - D_k(x_k)] &= \mathbb{E}[F(x_k)] - F^* + F^* - \mathbb{E}[D_k(x_k)] \\
&\leq \frac{1}{\tau}(1-\tau)^{k+1}(F^* - D_0(x_0) + (1-\tau)^k(F^* - D_0(x_0)) \\
&= \frac{1}{\tau}(1-\tau)^k(F^* - D_0(x_0)).
\end{aligned}$$

$\square$

### D.3 Accelerating MISO-Prox

The convergence rate of MISO (or also SDCA) requires a special handling since it does not satisfy exactly the condition (8) from Proposition 3.2. The rate of convergence is linear, but with a constant proportional to $F^* - D_0(x_0)$ instead of $F(x_0) - F^*$ for many classical gradient-based approaches.

To achieve acceleration, we show in this section how to obtain similar guarantees as Proposition 3.2 and 3.4—that is, how to solve efficiently the subproblems (5). This essentially requires the right initialization each time MISO-Prox is called. By initialization, we mean initializing the variables $z_i^0$.

Assume that MISO-Prox is used to obtain $x_{k-1}$ from Algorithm 1 with $G_{k-1}(x_{k-1}) - G_k^* \leq \varepsilon_{k-1}$, and that one wishes to use MISO-Prox again on $G_k$ to compute $x_k$. Then, let us call $D'$ the lower-bound of $G_{k-1}$ produced by MISO-Prox when computing $x_{k-1}$ such that

$$x_{k-1} = \arg\min_{x\in\mathbb{R}^p}\left\{D'(x) = \frac{1}{n}\sum_{i=1}^{n} d_i'(x) + \psi(x)\right\},$$

with

$$d_i'(x) = \frac{\mu + \kappa}{2}\|x - z_i'\|^2 + c_i'.$$

Note that we do not index these quantities with $k-1$ or $k$ for the sake of simplicity. The convergence of MISO-Prox may ensure that not only do we have $G_{k-1}(x_{k-1}) - G_k^* \leq \varepsilon_{k-1}$, but in fact we have the stronger condition $G_{k-1}(x_{k-1}) - D'(x_{k-1}) \leq \varepsilon_{k-1}$. Remember now that

$$G_k(x) = G_{k-1}(x) + \frac{\kappa}{2}\|x - y_{k-1}\|^2 - \frac{\kappa}{2}\|x - y_{k-2}\|^2,$$

and that $D'$ is a lower-bound of $G_{k-1}$. Then, we may set for all $i$ in $\{1, \ldots, n\}$

$$d_i^0(x) = d_i'(x) + \frac{\kappa}{2}\|x - y_{k-1}\|^2 - \frac{\kappa}{2}\|x - y_{k-2}\|^2,$$

which is equivalent to initializing the new instance of MISO-Prox with

$$z_i^0 = z_i' + \frac{\kappa}{\kappa + \mu}(y_{k-1} - y_{k-2}),$$

and by choosing appropriate quantities $c_i^0$. Then, the following function is a lower bound of $G_k$

$$D_0(x) = \frac{1}{n}\sum_{i=1}^n d_i^0(x) + \psi(x).$$

and the new instance of MISO-Prox to minimize $G_k$ and compute $x_k$ will produce iterates, whose first point, which we call $x^0$, minimizes $D_0$. This leads to the relation

$$x^0 = \mathrm{prox}_{\psi/(\kappa+\mu)}\left[\bar{z}^0\right] = \mathrm{prox}_{\psi/(\kappa+\mu)}\left[\bar{z}' + \frac{\kappa}{\kappa + \mu}(y_{k-1} - y_{k-2})\right],$$

where we use the notation $\bar{z}^0 = \frac{1}{n}\sum_{i=1}^n z_i^0$ and $\bar{z}' = \frac{1}{n}\sum_{i=1}^n z_i'$ as in Algorithm 2.

Then, it remains to show that the quantity $G_k^* - D_0(x^0)$ is upper bounded in a similar fashion as $G_k(x_{k-1}) - G_k^*$ in Propositions 3.2 and 3.4 to obtain a similar result for MISO-Prox and control the number of inner-iterations. This is indeed the case, as stated in the next lemma.

**Lemma D.5** (**Controlling** $G_k(x_{k-1}) - G_k^*$ **for MISO-Prox).**
*When initializing MISO-Prox as described above, we have*

$$G_k^* - D_0(x^0) \leq \varepsilon_{k-1} + \frac{\kappa^2}{2(\kappa + \mu)}\|y_{k-1} - y_{k-2}\|^2.$$

*Proof.* By strong convexity, we have

$$D_0(x^0) + \frac{\kappa}{2}\|x^0 - y_{k-2}\|^2 - \frac{\kappa}{2}\|x^0 - y_{k-1}\|^2 = D_0'(x^0) \geq D_0'(x_{k-1}) + \frac{\kappa + \mu}{2}\|x^0 - x_{k-1}\|^2.$$

Consequently,

$$\begin{aligned}
D_0(x^0) &\geq D'(x_{k-1}) - \frac{\kappa}{2}\|x^0 - y_{k-2}\|^2 + \frac{\kappa}{2}\|x^0 - y_{k-1}\|^2 + \frac{\kappa + \mu}{2}\|x^0 - x_{k-1}\|^2 \\
&= D_0(x_{k-1}) + \frac{\kappa}{2}\|x_{k-1} - y_{k-2}\|^2 - \frac{\kappa}{2}\|x_{k-1} - y_{k-1}\|^2 - \frac{\kappa}{2}\|x^0 - y_{k-2}\|^2 + \frac{\kappa}{2}\|x^0 - y_{k-1}\|^2 \\
&\quad + \frac{\kappa + \mu}{2}\|x^0 - x_{k-1}\|^2 \\
&= D_0(x_{k-1}) - \kappa\langle x^0 - x_{k-1}, y_{k-1} - y_{k-2}\rangle + \frac{\kappa + \mu}{2}\|x^0 - x_{k-1}\|^2 \\
&\geq D_0(x_{k-1}) - \frac{\kappa^2}{2(\kappa + \mu)}\|y_{k-1} - y_{k-2}\|^2,
\end{aligned}$$

where the last inequality is using a simple relation $\frac{1}{2}\|a\|^2 + 2\langle a, b\rangle + \frac{1}{2}\|b\|^2 \geq 0$. As a result,

$$\begin{aligned}
G_k^* - D_0(x^0) &\leq G_k^* - D_0(x_{k-1}) + \frac{\kappa^2}{2(\kappa + \mu)}\|y_{k-1} - y_{k-2}\|^2 \\
&\leq G_k(x_{k-1}) - D_0(x_{k-1}) + \frac{\kappa^2}{2(\kappa + \mu)}\|y_{k-1} - y_{k-2}\|^2 \\
&= G_{k-1}(x_{k-1}) - D'(x_{k-1}) + \frac{\kappa^2}{2(\kappa + \mu)}\|y_{k-1} - y_{k-2}\|^2 \\
&\leq \varepsilon_{k-1} + \frac{\kappa^2}{2(\kappa + \mu)}\|y_{k-1} - y_{k-2}\|^2
\end{aligned}$$

$\square$

We remark that this bound is half of the bound shown in (32). Hence, a similar argument gives the bound on the number of inner iterations. We may finally compute the iteration-complexity of accelerated MISO-Prox.

**Proposition D.6** (Iteration-Complexity of Accelerated MISO-Prox).
*When $F$ is $\mu$-strongly convex, the accelerated MISO-Prox algorithm achieves the accuracy $\varepsilon$ with an expected number of iteration upper bounded by*

$$O\left(\min\left\{\frac{L}{\mu}, \sqrt{\frac{nL}{\mu}}\right\}\log\left(\frac{1}{\varepsilon}\right)\log\left(\frac{L}{\mu}\right)\right).$$

*Proof.* When $n > 2(L-\mu)/\mu$, there is no acceleration. The optimal value for $\kappa$ is zero, and we may use Theorem 4.1 and Lemma C.1 to obtain the complexity

$$O\left(\frac{L}{\mu}\log\left(\frac{L}{\mu}\frac{F(x_0) - D_0(x_0)}{\varepsilon}\right)\right).$$

When $n < 2(L-\mu)/\mu$, there is an acceleration, with $\kappa = 2(L-\mu)/\mu - \mu$. Let us compute the global complexity using the "template" presented in Appendix C. The number of outer iteration is given by

$$k_{\text{out}} = O\left(\sqrt{\frac{L}{n\mu}}\log\left(\frac{F(x_0) - F^*}{\varepsilon}\right)\right).$$

At each inner iteration, we initialize with the value $x^0$ described above, and we use Lemma D.5:

$$G_k^* - D_0(x^0) \le \varepsilon_{k-1} + \frac{\kappa}{2}\|y_{k-1} - y_{k-2}\|^2.$$

Then,

$$\frac{G_k^* - D_0(x^0)}{\varepsilon_k} \le \frac{R}{2},$$

where

$$R = \frac{2}{1-\rho} + \frac{2592\kappa}{\mu(1-\rho)^2(\sqrt{q}-\rho)^2} = O\left(\left(\frac{L}{n\mu}\right)^2\right).$$

With Miso-Prox, with have $\tau_{G_k} = \frac{1}{2n}$, thus the expected number of inner iteration is given by Lemma C.1:

$$k_{\text{in}} = O(n\log(n^2 R)) = O\left(n\log\left(\frac{L}{\mu}\right)\right).$$

As a result,

$$\text{Comp} = O\left(\sqrt{\frac{nL}{\mu}}\log\left(\frac{F(x_0) - F^*}{\varepsilon}\right)\log\left(\frac{L}{\mu}\right)\right).$$

To conclude, the complexity of the accelerated algorithm is given by

$$O\left(\min\left\{\frac{L}{\mu}, \sqrt{\frac{nL}{\mu}}\right\}\log\left(\frac{1}{\varepsilon}\right)\log\left(\frac{L}{\mu}\right)\right).$$

$\square$

# E    Implementation Details of Experiments

In the experimental section, we compare the performance with and without acceleration for three algorithms SAG, SAGA and MISO-Prox on $l_2$-logistic regression problem. In this part, we clarify some details about the implementation of the experiments.

Firstly, we normalize the observed data before running the regression. Then we apply Catalyst using parameters according to the theoretical settings. Standard analysis of the logistic function shows that the Lipschitz gradient parameter $L$ is $1/4$ and strongly convex parameter $\mu = 0$ when there is no

regularization. Adding properly a $l_2$ term generates the strongly-convex regimes. Several parameters need to be fixed at the beginning stage. The parameter $\kappa$ is set to its optimal value suggested by theory, which only depends on $n$, $\mu$ and $L$. More precisely, $\kappa$ writes as $\kappa = a(L - \mu)/(n + b) - \mu$, with $(a, b) = (2, -2)$ for SAG, $(a, b) = (1/2, 1/2)$ for SAGA and $(a, b) = (1, 1)$ for MISO-Prox. The parameter $\alpha_0$ is initialized as the positive solution of $x^2 + (1 - q)x - 1 = 0$ where $q = \sqrt{\mu/(\mu + \kappa)}$. Furthermore, since the objective function is always positive, $F(x_0) - F^*$ can be upper bounded by $F(x_0)$ which allow us to set the $\varepsilon_k = (2/9)F(x_0)(1 - \rho)^k$ in the strongly convex case and $\varepsilon_k = 2F(x_0)/9(k + 2)^{4+\eta}$ in the non-strongly convex case. Finally, we set the free parameter in the expression of $\varepsilon_k$ as follows. We simply set $\rho = 0.9\sqrt{q}$ in the strongly convex case and $\eta = 0.1$ in the non strongly convex case.

To solve the subproblem at each iteration, the step-sizes parameter for SAG, SAGA and MISO are set to the values suggested by theory, which only depend on $\mu$, $L$ and $\kappa$. All of the methods we compare store $n$ gradients evaluated at previous iterates of the algorithm. For MISO, the convergence analysis in Appendix D leads to the initialization $x_{k-1} + \frac{\kappa}{\mu+\kappa}(y_{k-1} - y_{k-2})$ that moves $x_{k-1}$ closer to $y_{k-1}$ and further away from $y_{k-2}$. We found that using this initial point for SAGA was giving slightly better results than $x_{k-1}$.