[Reviews · NeurIPS 2015]

Submitted by Assigned_Reviewer_1

The paper definitely contains some new contributions from a technical point of view. However, the main idea and the theoretical results are not mature enough. Indeed, Algorithm 1 can be viewed as a simple inexact variant of Guler's fast proximal method in [9], where the authors impose the condition (5). This approach has several limitations including the choice of \kappa, the solution of the subproblem in (5).

The authors provide two convergence theorems for both the strongly convex case and the nonstrongly case. Unfortunately, the choice of the inner accuracy \epsilon_k depends on the optimal value F* which is unknown. In addition, the method for solving (5) is elusive and its convergence guarantee is given by \tildle{O} notion, which does not precisely reflect the real number of iterations of the inner loop. Hence, the overall complexity may be worse than existing methods. In addition, the convergence guarantees in (7) and (11) remain depending on F(x^0) - F*, which is different from existing methods.

As far as I observed, the authors fix kappa in Algorithm 1, but this is not a good idea since in the accelerated scheme, the error of the inexact oracle

accumulates. Indeed, the closer we are to the true solution the more exact solution of the subproblem we require. Hence, adaptively update kappa is crucial.

Clearly, extending Algorithm 1 from noncomposite to composite form has in issues, especially when the regularizer does not have a low cost proximal operator. The inner loop requires such an operator at each iteration.

Overall, even though the paper contains some new technical contributions, but they are not sufficiently strong. In addition, the convergence analysis has not done rigorously. I think to overcome this limitation, one can exploit the idea of sliding gradient/conditional gradient methods from G. Lan to characterize the convergence of Algorithm 1.
Summary: This paper proposes a so-called catalyst algorithm for unconstrained convex optimization. In fact, this algorithm can be viewed as an inexact variant of fast proximal method introduced by Guler in [9]. The authors analyze the convergence of their algorithm for two cases: strongly convex and nonstrongly convex. Several extensions and modifications are mentioned but are not rigorously backed up. A few numerical experiments are given to show advantages of their method.

Submitted by Assigned_Reviewer_2

I think this is a very interesting and significant piece of work, with several relevant and non-trivial contributions. Not only does the paper introduce a generic scheme to accelerate several optimization methods (namely several recent incremental methods), but it also extends the domain of applicability of some of those methods. The paper is very well written and the contributions are carefully put in the context of the state of the art. The experimental results, although naturally limited by the scarcity of space, are clear and convincing. As far as this reviewer knows, the results are original.

It has been recently shown that the convergence rate of Nesterov-type acceleration methods can be significantly boosted by using adaptive restart techniques. See for example, the paper "Adaptive Restart for Accelerated Gradient Schemes", by O'Donoghue and Candes, and the paper "Monotonicity and Restart in Fast Gradient Methods", by Giselsson and Boyd. It would be interesting to consider if the method proposed in this manuscript could also benefit from some form of restarting scheme, given that it also uses Nesterov-type extrapolation.

Summary: I think this is a very interesting and relevant piece of work, which contains several important and non-trivial contributions (see details below).

Submitted by Assigned_Reviewer_3

As mentioned in the paper, the algorithm in the paper is basically the same as in [9]. It combines extrapolation idea from Nesterov's acceleration scheme with proximal point operator. The algorithm in [9] used exact minimization in each iteration, which is not possible in practice. The current paper carries out the analysis with more realistic assumptions of approximate minimization. Paper [8] is quite similar in the above respects. It's stated that the current paper is independent of [8], but given that [8] has already been published, in my view the current paper should be considered only on the basis of what it has in addition to the results that overlap with [8]. There is a detailed comparison of the results here and in [8] in the appendix, but I would have also liked to see a comparison of the methods. Thus one of the main new result in the current paper is that it can also handle functions that are not strongly convex. But it seems to me---but I could be wrong as I didn't actually read the proofs---that the methods of proof are similar for the strongly convex and general convex cases. So I guess the question to answer is: Does the method of [8] also easily give the case where the function is not required to be strongly convex or is a new idea needed?

The other additional result is the acceleration and other improvements of Finito/MISO. The modified algorithm here turns out to be very similar to a variant of SDCA [24], though there are differences. Given this, the experimental results should have included a comparison with SDCA, and I consider this a major omission.
Summary: The paper provides an analysis of a generic acceleration method for first-order optimization based on Nesterov's extrapolation idea and proximal point operator.

Submitted by Assigned_Reviewer_4

This paper essentially extends Neterov's first acceleration technique for proximal point algorithms, considering inexact solutions of subproblems. Nontrivial modifications to exiting analyses were made to allow for controlled inexactness, so that global convergence rate can be derived depending on it.

The paper presents technical ideas with clarity and high quality. As far as I know, this work is original, which brings a new unified framework and understanding for accelerating first order methods with inexact inner solvers.

Personally, it was a bit disappointing to see in Thm 3.3 that the amount of inexactness has to decrease in a rapid rate as global iterations go on, but it is also expected.

Summary: This paper provides a unified acceleration framework and analysis for first-order optimization methods, considering carefully of inexactness of solving subproblems and showing its effect on convergence rate. Although I didn't check all the proofs, the theory looks solid and provides deeper understanding of the acclaimed acceleration techniques.

Submitted by Assigned_Reviewer_5

The authors propose a general scheme of wrapping an existing first order gradient method to solve the minimization of the original objective plus an iteratively sharper proximal term. The authors show that for a generic first order method this 'accelerates' the convergence by changes the depends on \frac{L}{\mu} to \sqrt{\frac{L}{\mu}}.

Overall, it is a goodpaper with a couple deficiencies in

(a) experiments and

(b) given the existing algorithms, when exactly would a catalyst be useful ? (especially in ML).

Despite the deficiencies, I still vote for acceptance as it has some ideas which could spur future research, for e.g. the idea of doing multiple passes over data with each pass having a sharper proximal term (similar to [8])

Comments

Page 2, line 57 - are the authors sure that 'all' the incremental methods have a O(n) cost ? Including ProxSVRG ?

The authors need to make it clear when their contributions are useful. For e.g., - when users cannot use the dual, and - ideally when the objective is not strongly convex (saying that n < L/mu seems, in my opinion, is a little weak and less useful regime)

- and when data cannot be held in memory (as the inner algorithm can make a sequential passes over data).

line 314, redundant 'do'

line 390 - what does 'default' parameters ? default as in optimal w.r.t theory ? or just happen to be parameters in some source code ? The authors of SAG also outlined a procedure to do learning rate scheduling - was this taken into account.

Was 'L' just set to the upperbound 'L' in the dataset ? was there any effort to tune this ?

How was \eta set ? Was there any effort taken to ensure that enough tuning was performed on all the algorithms ? or was it the case that the authors picked a set of values and lucked out by getting a better performance on the proposed scheme ?

One of the benefits of using a method like SAG/SAGA/SVRG/.. is to ensure quick initial convergence which is not guaranteed by FG or Acc-FG. It would be nice to see how the proposed method does w.r.t Acc-FG.

Suggestions

Theorem 3.3. is misleading at a first glance, could the authors please ensure the contents of line line 256-257 are located in a more accessible place ? like the theorem statement itself.

Table 1 caption " To simplify, we only present the case where n <= L/\mu when \mu > 0." could be boldened to ensure that it is read
Summary: The authors propose a general scheme of wrapping an existing first order gradient method to solve the minimization of the original objective plus an iteratively sharper proximal term. The authors show that for a generic first order method this 'accelerates' the convergence by changes the depends on \frac{L}{\mu} to \sqrt{\frac{L}{\mu}}. Despite multiple shortcomings, I think this work passes the bar for acceptance as it has interesting ideas that could spur future research.

Author Feedback
Author rebuttal: We thank the reviewers for their constructive comments and respond below to their comments.

** Main message of the paper **
a) Discussing the useful regimes (R3)
We thank the reviewer for his advice, which we will follow, if it is accepted. We will also take into account his suggestions regarding Th 3.3 and Table 1.

b) Alg1 is a meta-algorithm (R4)
We respectfully disagree with several comments made by R4. Alg1 is a meta-algorithm that can be applied to a large class of first-order methods. It is thus normal not to precise in Sections 2 and 3 which method to use for solving (5). Similarly, we do not see any ``issue'' with composite optimization. Some methods may require to compute the proximal operator of psi. This is feasible for a specific class of functions, which is classical in the literature about composite optimization.

c) Novelty wrt Nesterov (R6)
Nesterov's acceleration was originally proposed for the full gradient method, and extended later to some other methods. For several methods, such as SAG, SAGA, Finito/MISO, the extension was still an open problem.

** Technical points **
a) use of adaptive restart techniques (R1)
We thank the reviewer for this promising suggestion, which we will investigate in a journal version of this paper.

b) O(n) storage cost (R3)
The statement is true for most methods we cite, but the reviewer is right that for Prox-SVRG, the storage of gradients O(n) can be replaced by that of one iterate O(p) and extra gradient computations. We thank the reviewer for his remark and will replace our statement by a more accurate one.

c) Notation \tilde{O} (R4)
Our results are essentially non-asymptotic thanks to the use of estimate sequence, and the notation \tilde{O} is only used to make the paper more readable. The exact constants follow from the calculations detailed in Sec. C of the appendix.

d) epsilon_k depends on F* (R4)
The quantity F(x_0)-F* can be replaced by any upper-bound, as stated in lines 210-214, which does not break the theory and makes the algorithm practical.

e) convergence rate depends on F(x^0)-F* (R4)
This is a classical quantity that appears often in convergence rates (see e.g., SAG, SAGA). Thus, we did not understand the concern of the reviewer.

f) fixing kappa is ``not a good idea'' (R4)
We choose kappa according to the theory, leading to a fixed value that depends on mu and L. Using a variable kappa may be an interesting direction to study, but there is so far neither theoretical nor empirical evidence that there exists a simple way to do that.

g) use of sliding gradient (R4)
We did not understand the relation between the sliding gradient and our work. Instead, we are planning to cite a recent arxiv paper of G. Lan called ``An optimal randomized primal-dual gradient method'', which is related to our work.

h) Proofs for convex vs strongly convex (R5)
To the best of our knowledge, the analysis of [8] cannot be extended to non-strongly convex functions in a straightforward way. One key of the analysis of [8] is to handle the error term by cancelling a part of the strongly convex term at each iteration (a technique introduced in [25]). In contrast to [8], our analysis controls the error accumulation related to those terms, leading to a more general result that does not require strong convexity, and with a constant improved by a factor 2. This key novel steps are (i) introducing the right estimate sequence, (ii) the right definition of approximate estimate sequence, and (iii) the right analysis for controlling the error accumulation.

** Experimental part **
a) Scope of the experiment (R3, R5)
Because of space limitation, we have chosen a single ``proof-of-concept'' experiment to show that catalyst is a generic purpose acceleration scheme. With this goal in mind, we have applied catalyst to three methods that did not admit any existing accelerated variant (SAG, Finito/Miso and SAGA). We agree that there are other interesting experiments to perform to fully understand the empirical behavior of catalyst, which we are planning to do for a journal version of our paper. This includes: comparison with FG/Acc-FG, acceleration of other inner-loop algorithms (BCD), use of more datasets...

b) Experimental setup (R3)

In all experiments, no adaptive tuning was used. L was set to the upper bound, and \mu to the lower bound. According to theory, the optimal choice of \kappa only depends on these parameters. The value of \eta needs to be small (10) while keeping the constant in (11) reasonable. 0.1 was thus a natural value for that and no further tuning was used. We also check what happens when \eta=0.01 and we obtain very similar results. For other approaches, we also follow recommendations from original papers, e.g., we use the stepsize 1/L for SAG, which does not produce dramatically worse results than the heuristic SAG-LS (see Fig 3 in [23]). We will clarify the experimental setup in the paper, if it is accepted.